# Structure and pro-toxic mechanism of the human Hsp90/PPIase/Tau complex

Javier Oroz[1,8], Bliss J. Chang[2], Piotr Wysoczanski[1,2], Chung-Tien Lee[3], Ángel Pérez-Lara[4],
Pijush Chakraborty[1], Romina V. Hofele[3], Jeremy D. Baker[5], Laura J. Blair[5], Jacek Biernat[6], Henning Urlaub[3,7],
Eckhard Mandelkow[6], Chad A. Dickey[5] & Markus Zweckstetter[1,2]

The molecular chaperone Hsp90 is critical for the maintenance of cellular homeostasis and represents a promising drug target. Despite increasing knowledge on the structure of Hsp90, the molecular basis of substrate recognition and pro-folding by Hsp90/co-chaperone complexes remains unknown. Here, we report the solution structures of human full-length Hsp90 in complex with the PPIase FKBP51, as well as the 280 kDa Hsp90/FKBP51 complex bound to the Alzheimer's disease-related protein Tau. We reveal that the FKBP51/Hsp90 complex, which synergizes to promote toxic Tau oligomers in vivo, is highly dynamic and stabilizes the extended conformation of the Hsp90 dimer resulting in decreased Hsp90 ATPase activity. Within the ternary Hsp90/FKBP51/Tau complex, Hsp90 serves as a scaffold that traps the PPIase and nucleates multiple conformations of Tau's proline-rich region next to the PPIase catalytic pocket in a phosphorylation-dependent manner. Our study defines a conceptual model for dynamic Hsp90/co-chaperone/client recognition.

[1] German Center for Neurodegenerative Diseases (DZNE), Von-Siebold-Straße 3a, 37075 Göttingen, Germany. [2] Department for NMR-based Structural Biology, Max Planck Institute for Biophysical Chemistry, Am Faßberg 11, 37077 Göttingen, Germany. [3] Bioanalytical Mass Spectrometry, Max Planck Institute for Biophysical Chemistry, Am Faßberg 11, 37077 Göttingen, Germany. [4] Department of Neurobiology, Max Planck Institute for Biophysical Chemistry, Am Faßberg 11, 37077 Göttingen, Germany. [5] Department of Molecular Medicine, Morsani College of Medicine, USF Health Byrd Alzheimer's Institute, University of South Florida, Tampa, FL 33613, USA. [6] DZNE, CAESAR Research Center, Ludwig-Erhard-Alle 2, 53175 Bonn, Germany. [7] Bioanalytics Group, Institute for Clinical Chemistry, University Medical Center, Robert-Koch-Straße 40, 37075 Göttingen, Germany. [8] Present address: Instituto de Química-Física Rocasolano, IQFR-CSIC, Serrano 119, 28006 Madrid, Spain. Correspondence and requests for materials should be addressed to M.Z. (email: Markus.Zweckstetter@dzne.de). Deceased: Chung-Tien Lee, Chad A. Dickey.

The molecular chaperone heat-shock protein 90 (Hsp90) is evolutionarily conserved and highly abundant in the cytosol of eukaryotic cells where, in cooperation with a large number of (co)-chaperones, it is responsible for stabilization, maturation, regulation, and activation of a wide range of client proteins[1–6]. Despite an increasing knowledge on the interaction of Hsp90 with different clients, little is still known about the recognition mechanisms that rule these interactions[7], or the cooperative effect of co-chaperones on client recognition and pro-folding action[5,8]. Because chaperones select proteins for refolding or degradation, they determine the progression of misfolding diseases, where the aberrant folding of specific proteins promotes a toxic gain of function[1,9,10]. The most common misfolding disease is Alzheimer's disease (AD), which is hallmarked by the oligomeric aggregation and amyloid fibril formation of amyloid-β and the microtubule-associated protein Tau[11]. Hsp90 promotes either degradation or toxic aggregation of its natively disordered client Tau[12], depending on the particular associated co-chaperone[13].

Interaction of Hsp90 with the *cis–trans* peptidyl-prolyl isomerase (PPIase) FKBP51 (FK506-binding protein of 51 kDa)[14] promotes amorphous aggregation of Tau[15–17], and drives neuronal death in vivo[17,18]. In addition, expression levels of the Hsp90 pro-folding co-chaperone FKBP51 increase with age, are elevated in AD brain, and correlate with Tau pathogenesis[17,18]. Insights into the structural changes that depend on the Hsp90/FKBP51 pro-toxic complex and trigger Tau aggregation and toxicity are thus important to decipher the nature of Tau's toxic pathway.

The way chaperones accomplish their functions is a matter of intense debate in structural biology[19–21]. The structures of several Hsp90/co-chaperone complexes were solved[22,23], some including a client protein[19,24]. The structural basis of Hsp90/PPIase complexes, however, has remained enigmatic, because of the dynamic nature of the interaction[5]. Using an integrative approach that combines functional assays with fluorescence-based and calorimetric binding studies, mutational analysis, nuclear magnetic resonance (NMR) spectroscopy, and chemical cross-linking (XL), we determined the solution structures of the human Hsp90/FKBP51 complex and the Hsp90/FKBP51 complex bound to its intrinsically disordered substrate Tau. Our study reveals that FKBP51 binds in between the two arms of the Hsp90 dimer, in a manner that stabilizes Hsp90's open conformation and decreases its ability to hydrolyze ATP. Within the Hsp90/FKBP51/substrate complex, Hsp90 serves as a scaffold that traps the PPIase and nucleates multiple conformations of Tau's proline-rich region next to the PPIase catalytic pocket, providing a potential basis for the pro-toxic role of the Hsp90/FKBP51 complex. A key finding of our study is that the interaction of the PPIase with Hsp90 is dynamic and involves Hsp90 surfaces distinct from other co-chaperones, thus enabling asymmetric chaperone/co-chaperone/client complexes.

## Results

**Hsp90 interacts with FKBP51 in an open conformation.** The biological activity of the abundant chaperone Hsp90 depends on a tightly regulated equilibrium of extended and closed dimeric molecular conformations[5,25,26] (Fig. 1a). Small-angle X-ray scattering (SAXS) showed that human Hsp90 populates predominantly extended states (Fig. 1b and Supplementary Figure 1a, b)[26,27]. In the presence of the PPIase FKBP51, which binds with high affinity and equimolar stoichiometry to the Hsp90 dimer[28], the overall dimensions further increased (Fig. 1b). The Hsp90/FKBP51 complex remained open in the presence of nucleotides (Supplementary Figure 1b), despite the

ability of nucleotides to promote compact Hsp90 conformations[25].

In addition, FKBP51, but not the intrinsically disordered substrate Tau[27], decreased the rate of Hsp90 ATP hydrolysis (Fig. 1c). Hsp90 ATPase activity was restored by the addition of Aha1, a strong enhancer of Hsp90 ATPase activity through compaction of the Hsp90 conformation (Fig. 1c)[5,22]. Taken together, the data show that FKBP51 stabilizes Hsp90 in an extended state, impeding the specific conformational changes required for ATP hydrolysis[25,29,30].

**FKBP51 binds to a large continuous Hsp90 surface.** To elucidate the structural basis of the Hsp90/FKBP51 interaction, we protonated Hsp90 at isoleucine methyl side chains in an otherwise deuterated protein to visualize these specific groups by two-dimensional (2D) methyl-transverse relaxation optimized spectroscopy (TROSY) NMR spectra (Fig. 1d and Supplementary Figure 2a)[27,31,32]. Upon addition of unlabeled, full-length human FKBP51, signal intensities of specific isoleucine groups were perturbed, consistent with binding occurring in the intermediate-to-slow exchange regime (Fig. 1d). Sequence-specific analysis of the NMR perturbations defined a large, continuous interface formed by all three Hsp90 domains (Fig. 1e and Supplementary Figure 2c). The interface includes the projecting loop (Fig. 1e), which is relevant for substrate interaction[30], as well as the loop between residues Leu385-Lys398, a primary determinant of Hsp90's conformational plasticity and ATPase activity[29,30].

Addition of nucleotides to the Hsp90/FKBP51 complex induced signal changes near the ATP-binding site in the Hsp90N domain (Fig. 1f and Supplementary Figure 1c–e), indicating that decreased ATPase activity in the presence of FKBP51 (Fig. 1c) is not caused by impaired nucleotide binding. Deletion of the C-terminal domain of Hsp90 furthermore impaired complex formation (Supplementary Figure 2d–e), consistent with the importance of the interaction between the C-terminal tail of Hsp90 and the tetratricopeptide repeats (TPRs) of FKBP51 (Fig. 1a)[33].

**Co-chaperone binding is dynamic.** Using several FKBP51 constructs and a large number of multidimensional NMR experiments, we also assigned all methyl groups of the FK1 and FK2 domains within the 52 kDa full-length FKBP51 (Fig. 2a and Supplementary Figures 1a and 2b). Consistent with the observations made for Hsp90 (Fig. 1d), signal intensities of specific FKBP51 methyl resonances were decreased upon addition of unlabeled Hsp90 (Fig. 2a–e and Supplementary Figure 2f). The binding-sensitive residues form a large interface and involve all three FKBP51 domains (Fig. 2f).

Binding studies at increasing Hsp90 concentrations showed that at equimolar Hsp90 concentration, NMR signals in FK1 and FK2 decreased on average by ~20% and ~25%, respectively (Fig. 2g), while residues in the TPR domain were attenuated by ~72% (Supplementary Figure 2g–h). At higher Hsp90 concentration, NMR signal intensities in FK1 and FK2 further decreased in parallel with the intensity decrease in the TPR domain (Fig. 2b–g).

The stepwise binding pattern indicates that FKBP51/Hsp90 complex formation is based on domain-specific contacts with affinities decreasing in the following order: TPR > FK2 > FK1 (Fig. 2e–g). Consistent with these observations, deletion of FK1 modestly affected the kinetics of binding to Hsp90, while removal of the TPR domain completely abolished complex formation (Fig. 2h). The data show that FK1 and FK2 domains populate an ensemble of partially bound and unbound conformations even at co-chaperone excess (Fig. 2i).

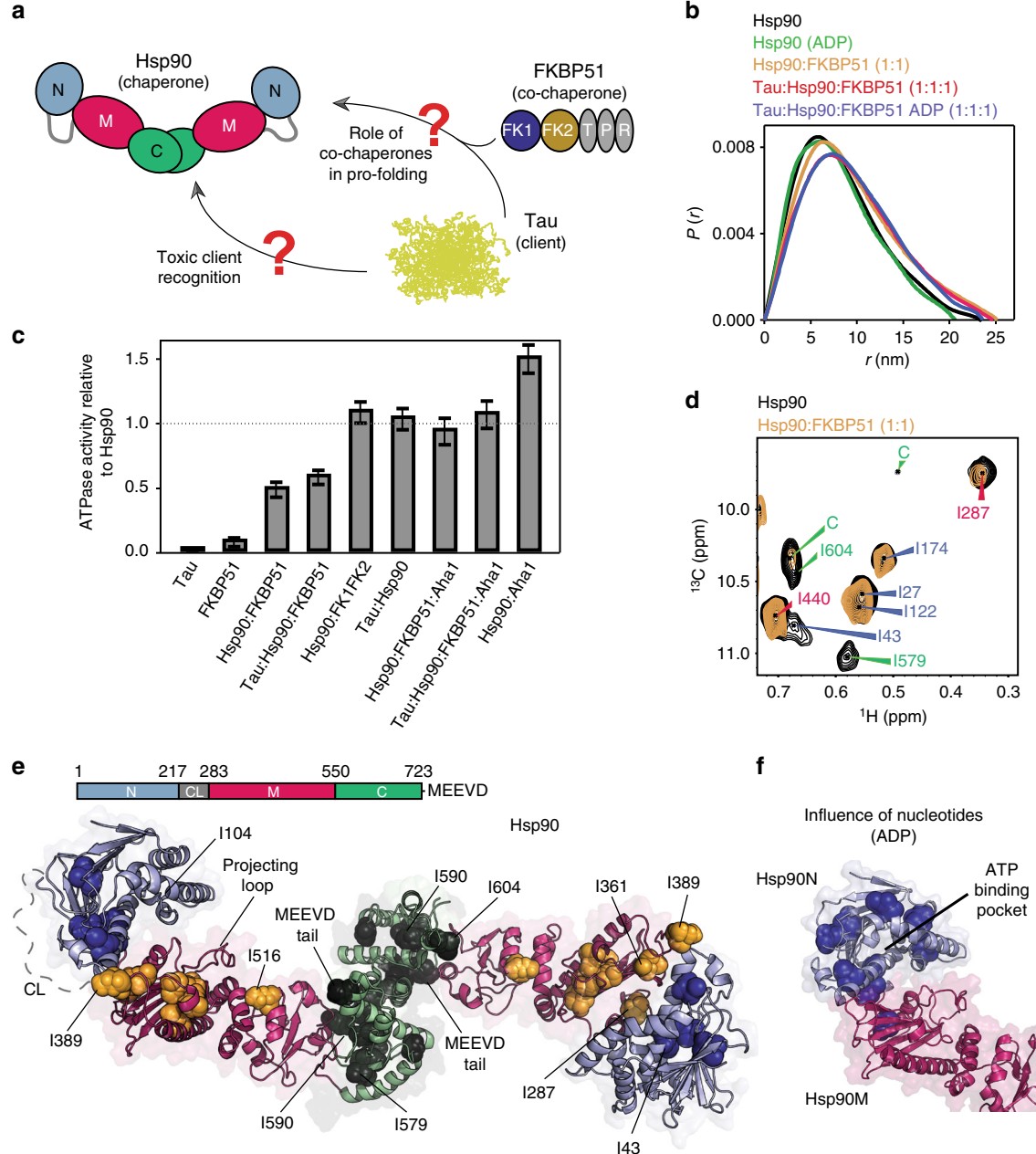

**Fig. 1** FKBP51 binds to a large continuous Hsp90 surface. **a** Schematic representation of the toxic Hsp90/FKBP51/Tau cross-talk. **b** $P(r)$ SAXS profiles show that FKBP51 and Tau bind to the extended conformation of Hsp90 dimer, even in the presence of nucleotides. **c** ATPase activity of Hsp90 in the presence of different co-chaperones and Tau. All measurements were performed in triplicate and error bars represent the standard error of the mean. **d** Methyl-TROSY spectra of Hsp90 (80 µM) in the absence (black) and presence of an equimolar ratio of unlabeled FKBP51 (orange). Two unassigned residues in the C-terminal domain of Hsp90 are marked by "C". **e** Structure of the extended human Hsp90β dimer (modified from PDB id: 5fwk[19]). Residues affected in FKBP51 titrations are represented by spheres. Selected residues are labeled. **f** Consistent with nucleotide binding to the N-terminal domain of Hsp90, ADP-induced changes (spheres) in NMR signal intensities were observed in proximity to the ATP-binding pocket in Hsp90N in the presence of saturating FKBP51 concentration

**Dissecting the dynamic 280 kDa Hsp90/FKBP51 complex.** To characterize the dynamic structure of the 280 kDa Hsp90/FKBP51 complex (Fig. 2i), we determined a complementary set of structural restraints (Supplementary Figure 3 and Supplementary Tables 1–2). 3D and 4D NOESY[34] experiments using nonlinear sampling techniques were recorded on isolated Ile-labeled Hsp90, on isolated Leu/Val-labeled FKBP51, on isolated Ala/Met-labeled FKBP51, and equimolar protein mixtures. In agreement with the 3D structures of the isolated proteins, many intramolecular NOEs

were identified (Supplementary Figure 4), indicating that the two proteins are properly folded in the complex.

The strength of the chosen labeling scheme is that inter-molecular NOEs between Hsp90 and FKBP51 are separated into distinct spectral regions, allowing identification of a few, non-overlapping and reliable NOEs between Hsp90 and FKBP51 (Supplementary Figure 3a and Supplementary Table 1). Using this strategy, seven intermolecular NOEs were assigned to the three domains of Hsp90 and FKBP51. They defined distances of

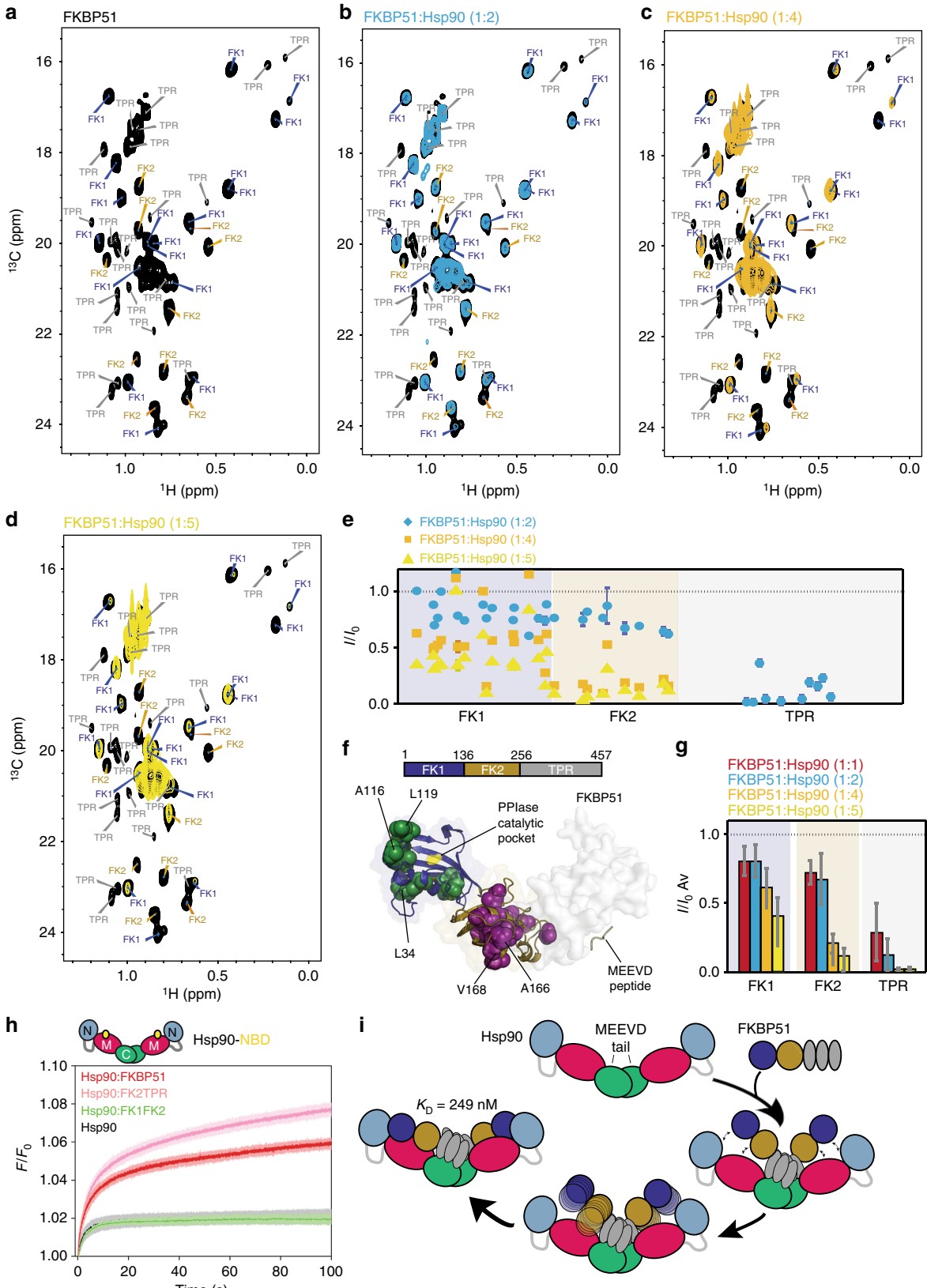

<~6 Å between individual methyl groups upon complex formation. Further evidence for direct contacts between Hsp90 and FKBP51 came from amide-directed chemical XL in combination with mass spectrometry (MS). Eleven robust XL-MS links were identified between the TPR domain of FKBP51 and Hsp90M/Hsp90C, one between FK2 and Hsp90C, three between FK1 and Hsp90M, and one between FK1 and Hsp90N (Supplementary Figure 3b and Supplementary Table 2).

**Fig. 2** Dynamic nature of the interaction of FKBP51 with Hsp90. **a–d** Methyl-TROSY spectra of leucine-valine labeled full-length human FKBP51 (100 μM) in the absence (black) and presence of increasing concentrations of unlabeled Hsp90. FKBP51:Hsp90 molar ratios are indicated (related to monomeric Hsp90). Although TPR residues were not sequence specifically assigned, isolated TPR cross-peaks can be identified through comparison with the complete assignment of FK1–FK2. **e** Residue-specific changes in the intensity of FKBP51 Leu/Val methyl groups observed in **a–d**. $I$ is the cross peak intensity of samples with FKBP51:Hsp90 molar ratios of 1:2 (blue), 1:4 (orange), and 1:5 (yellow). $I_0$ is the intensity of NMR signals of FKBP51 in the absence of Hsp90. Error bars were calculated based on NMR signal-to-noise ratio. TPR residues (Leu/Val) located at the MEEVD-binding site (PDB id: 5njx) were included into the analysis. **f** FKBP51 residues, which were strongly perturbed by Hsp90, are shown as spheres in the 3D structure (PDB id: 1kt0). The PPIase catalytic pocket in FK1 is marked in yellow. Selected residues in the FK1 and FK2 domains are labeled. **g** Domain-specific changes in NMR signal intensities of FKBP51 methyl groups observed at increasing Hsp90 concentrations. $I/I_0$ ratios were averaged over residues belonging to individual domains. Error bars represent SD. **h** Stopped-flow fluorescence-binding assays show that removal of FK1 (i.e., use of FK2-TPR) accelerates the rate of Hsp90 binding ($\Delta F_{FK2TPR} = 0.0027 \pm 0.0001\ s^{-1}$; $\Delta F_{FKBP51} = 0.0021 \pm 0.0003\ s^{-1}$). FK1–FK2, which lacks TPR repeats, showed no binding to Hsp90. **i** Schematic representation of domain-specific formation of the Hsp90/FKBP51 complex

**Structure of the Hsp90/FKBP51 complex.** Residue-specific signal attenuation of methyl groups (Figs. 1 and 2 and Supplementary Figure 2), domain–domain-specific interactions (Fig. 2), intermolecular NOEs (Supplementary Table 1), and XL-MS links (Supplementary Table 2) were subsequently combined into a structural model of the Hsp90/FKBP51 complex (Fig. 3a–c and Supplementary Table 6). The reliability of the structural model was validated by paramagnetic NMR (Fig. 3d–g): Hsp90 tagged with a S-(1-oxyl-2,2,5,5tetramethyl-2,5-dihydro-1H-pyrrol-3-yl) methyl methanesulfonothioate (MTSL) nitroxide tag at position 362 induced paramagnetic broadening of FK1 and FK2 residues, fully consistent with the 3D structure of the complex (Fig. 3d, e). In addition, binding of CLaNP-7-tagged FKBP51 to Hsp90 caused pseudo-contact shifts (PCSs) for selected methyl groups of Hsp90 (Fig. 3f, g). The PCS-affected isoleucines belong to Hsp90N and Hsp90M and are located within ~25 Å of the paramagnetic center in the structure of the Hsp90/FKBP51 complex (Fig. 3f).

The structure of the Hsp90/FKBP51 complex reveals an extended Hsp90 dimer (Fig. 3c), in agreement with the overall dimensions of the complex determined by SAXS (Fig. 1b). Within the complex, Hsp90 and FKBP51 are arranged in a head-to-head topology, with Hsp90N contacting FK1, Hsp90M binding to FK1/FK2/TPR, and Hsp90C interacting with FK2/TPR (Fig. 3b, c). The so-called projecting loop of Hsp90, which is important for stabilization of Hsp90 complexes[30], is located in between FK1 and FK2 (Fig. 3c). Hsp90N, which can move with respect to Hsp90M[35], is rotated to accommodate FK1 (Supplementary Figure 5a). This rotation positions the ATP-binding pocket away from R391, the critical residue for ATP hydrolysis[29], consistent with the observed decrease in Hsp90 ATPase activity upon binding to FKBP51 (Fig. 1c). Calorimetric analysis of the interaction shows a significant decrease in the enthalpy of binding for the Hsp90/FK2-TPR complex (Supplementary Figure 5b–c), indicating that the three domains of FKBP51 (Fig. 1a) contribute to form the large interaction surface on Hsp90 (Fig. 2i). The structural model of the Hsp90/FKBP51 complex further shows that the PPIase catalytic pocket located on FK1 is solvent accessible. In addition, the catalytic pocket is positioned close to the Hsp90 residue Gly324 and also to Hsp90's charged linker (Fig. 3a–c), which are both critical for binding to clients[36,37].

**Binding of the disordered Tau to the Hsp90/PPIase complex.** Next, we determined the molecular nature of the interaction of the intrinsically disordered protein Tau, which binds asymmetrically to the Hsp90 dimer[27], with Hsp90 and the Hsp90/FKBP51 complex, since this complex was the first neuronal chaperone complex to produce a toxic Tau intermediate[17]. Affinity measurements showed that FKBP51 has little influence on the micromolar affinity of Tau for human Hsp90[27] (Fig. 4a).

However, chromatographic measurements demonstrated that FKBP51 stabilizes the ternary complex (Fig. 4b, c), both in the case of full-length human Tau and a Tau fragment comprising the microtubule-binding region[16]. This corroborates previous findings that demonstrate FKBP51 stabilizes Tau[18]. Notably, the elution profile of the ternary complex showed a broad distribution of molecular weights (Fig. 4c), suggesting that Tau adopts a fuzzy arrangement in the complex[38], which has been previously shown to be spared from proteasomal degradation[18].

To identify Tau residues that interact with Hsp90, we performed NMR titrations using 15N-labeled, 441-residue Tau. According to the affinities for the Hsp90/FKBP51/Tau interaction (Fig. 4a), it is possible to saturate the ternary complex in NMR and SAXS measurements (Fig. 1b and Supplementary Figure 1b). Residue-specific analysis revealed that the proline-rich region and the four pseudo-repeats, which are part of Tau's microtubule-binding domain, bind to Hsp90 (Fig. 4d)[16,27]. Tau also interacted with lower affinity with FKBP51 (Fig. 4d and Supplementary Figure 6), which colocalizes with Tau in vivo[18], in the absence of Hsp90. The ability of FKBP51 to bind to Tau is in agreement with the PPIase's ability to interact with and refold misfolded clients independent of Hsp90[28].

**Paramagnetic NMR of the Tau interaction with Hsp90/FKBP51.** Tau's interface in Hsp90 and the Hsp90/FKBP51 complex was subsequently determined using paramagnetic NMR[39,40]. To this end, Tau was tagged with MTSL labels independently at three different positions located in the Hsp90-interacting region. Binding of the MTSL-tagged Tau protein to Hsp90 caused paramagnetic broadening in all three Hsp90 domains and defined a very large interaction surface (Fig. 5a, Supplementary Figure 7, and Supplementary Table 5). Binding of Tau to Hsp90 did not influence Hsp90's ATPase activity (Fig. 1c). Consistent with this observation, the Tau interaction surface is accessible in both the extended and closed Hsp90 dimer structures (Supplementary Figure 7–f).

Hsp90's surface for Tau-binding contains hydrophobic residues (Fig. 4e, f) that can interact with hydrophobic stretches within the microtubule-binding domain of Tau[16]. Moreover, phosphorylation of Tau by the microtubule affinity-regulating kinase 2[41] and insertion of glutamic acid at specific phosphorylation epitopes[42] abolished/weakened Tau's interaction with Hsp90 and FKBP51 (Fig. 4g, h), in agreement with in vivo results[43]. Because phosphorylation decreases Tau's net positive charge, the analysis shows that electrostatic interactions contribute to Hsp90/Tau multivalent complexation. Because phosphorylation of Tau is intimately connected to the progression of AD[16], the impaired interaction of phosphorylated Tau with Hsp90/FKBP51 suggests that the interplay of Tau with Hsp90/co-chaperone complexes may vary in different stages of AD.

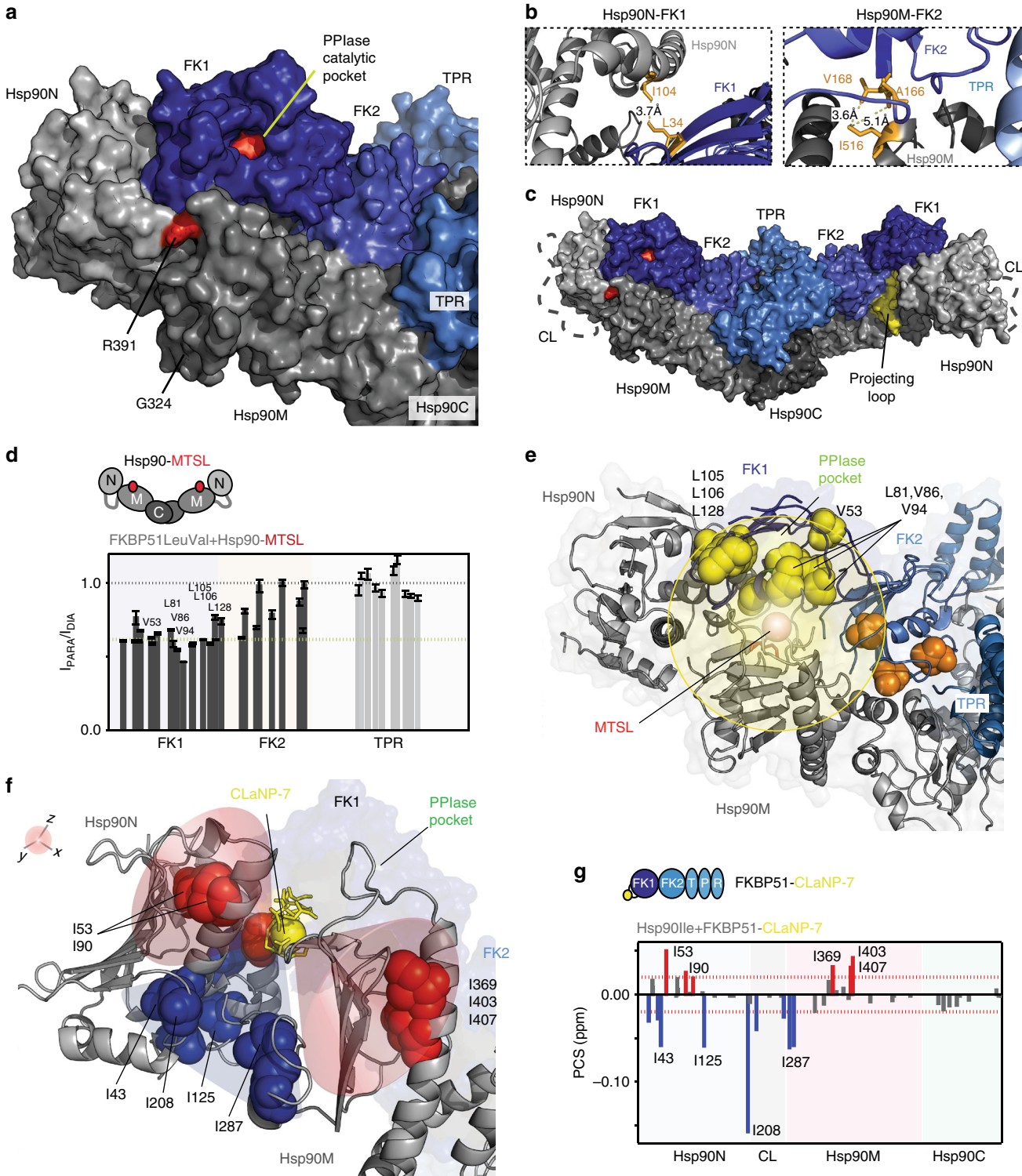

**Fig. 3** Head-to-head topology of the 280 kDa Hsp90/FKBP51 complex. **a–c** Structural model of the Hsp90/FKBP51 complex derived from a combination of structural and biochemical data. Hsp90 is colored gray, and FKBP51 in blue. Insets in **b** show intermolecular contacts, for which Hsp90/FKBP51-specific NOEs were detected. R391 (red) is critical for Hsp90's ATPase activity. The charged linker (CL) is represented as a gray dotted line in **c**, and Hsp90M's projecting loop is colored yellow. **d** PRE-induced signal broadening of FKBP51 moieties upon binding to Hsp90 tagged with MTSL at position 362. Intensity ratios for TPR cross-peaks are shown for comparison (gray), but were not sequence specifically assigned (i.e., placed at random positions in the plot within the MEEVD-binding region). Error bars were calculated based on NMR signal-to-noise ratio. Threshold (yellow line) was set to 0.6. **e** FKBP51 residues undergoing strongest PREs (below threshold in **d**) are represented in yellow spheres and are located in FK1 within 25 Å to the MTSL spin center, validating the structural model. Orange spheres mark FK2 residues with close-to-threshold PRE broadening (**d**). **f, g** PCSs experienced by Hsp90 moieties upon binding to FKBP51, which was labeled with a single CLaNP-7 lanthanide tag on FK1 (at positions C107–C110). The orientation of the lanthanide tensor (red and blue shadows in **f**) fits well with the experimental signs of PCSs (in red and blue spheres in **f** and bars in **g**). Red dotted lines in **g** represent 1.5 times the SD of the averaged chemical shift changes and are used as a threshold

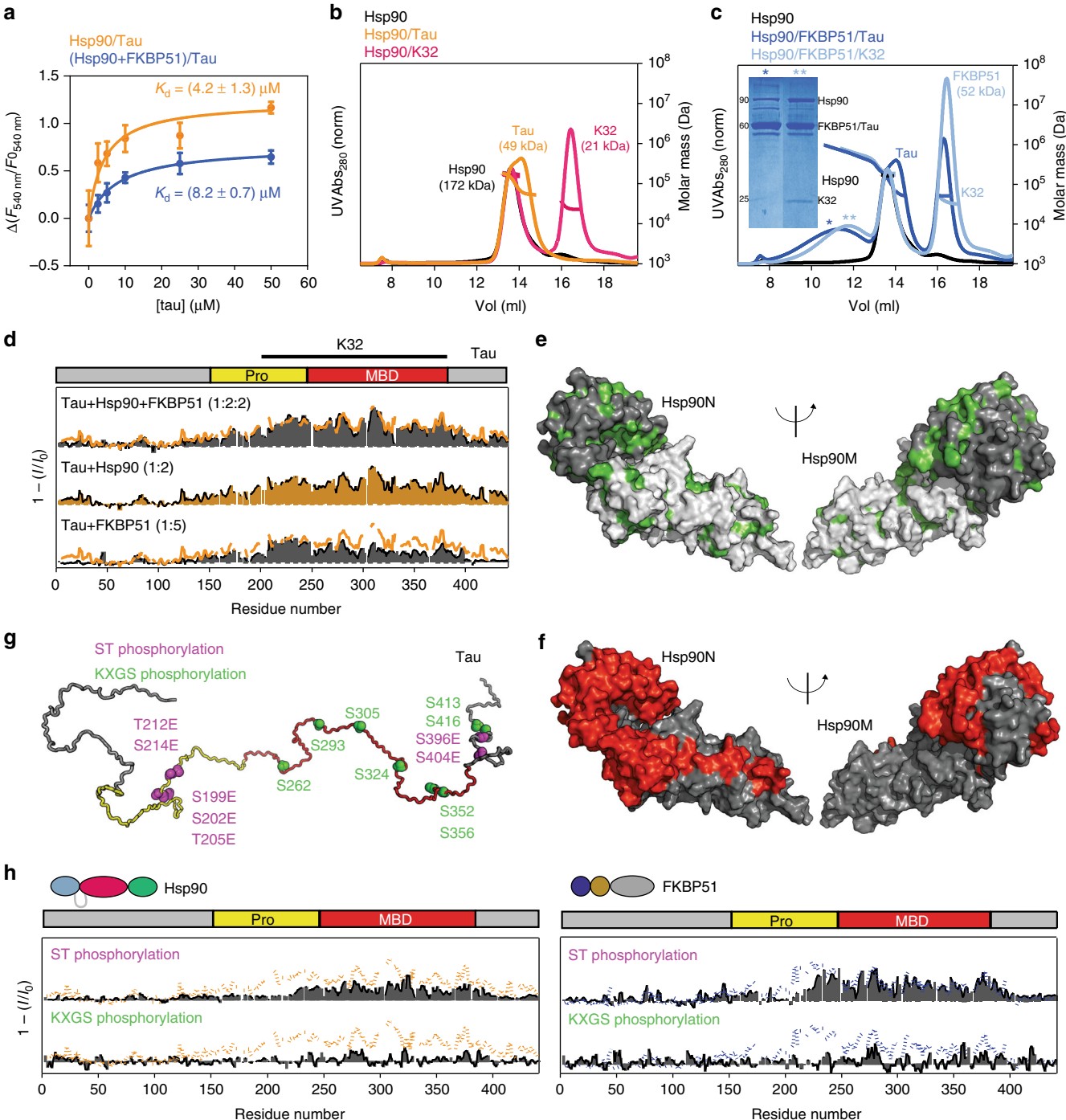

**Fig. 4** Phosphorylation-dependent complexation of Tau. **a** Binding affinity of Tau to Hsp90 is slightly decreased by the presence of FKBP51. Reported are average values of triplicate experiments ± SD. **b** The dynamic complex formed by Hsp90/Tau dissociates on the SEC column. **c** The presence of FKBP51 stabilizes the ternary complex. Fractions corresponding to the ternary complex (represented by asterisks) were submitted to SDS-PAGE showing the presence of the three proteins in the Hsp90/FKBP51/Tau and Hsp90/FKBP51/K32 ternary complexes. Note that FKBP51 and 441-residue Tau co-migrate in the gel resulting in overlapping bands on SDS-PAGE. Both full-length Tau and the K32 fragment[16] were used, to illustrate the reproducibility of the results. MALS fittings for molecular weight estimation are indicated. Note that Tau retention times are comparable to those of much larger proteins, due to its disordered structure. The uncropped gel is shown in Supplementary Figure 8. **d** NMR interaction profiles of 441-residue Tau with Hsp90 (orange; Tau/Hsp90 molar ratio of 1:2), FKBP51 (bottom; Tau/FKBP51 molar ratio of 1:5) and the Hsp90/FKBP51 complex (top). $I_0$ and $I$ are intensities of H–N cross-peaks of Tau in the absence and presence of the binding partner, respectively. For comparison, the binding pattern to Hsp90 is represented in the Tau/FKBP51 and Tau/Hsp90/FKBP51 plots by orange lines. The region covered by K32 fragment is shown on top. **e** Hydrophobic residues (green) in the N/M domains of Hsp90. **f** PREs induced in Hsp90 domains (red) by binding of Tau tagged with MTSL at position 256 within the binary Hsp90/Tau complex. **g** Phosphorylation of Tau impedes binding to Hsp90 and FKBP51. Cartoon representation indicating KXGS phosphorylation sites in Tau (green[41]), as well as sites for Ser-Thr pseudo-phosphorylation detected by AT8, AT100, and PHF1 antibodies (purple[42]). **h** NMR binding plots of Ser-Thr pseudo-phosphorylated (purple) and MARK2-phosphorylated (green) Tau to Hsp90 (left) and FKBP51 (right). For comparison, the binding profiles of wild-type Tau (dotted lines) are shown

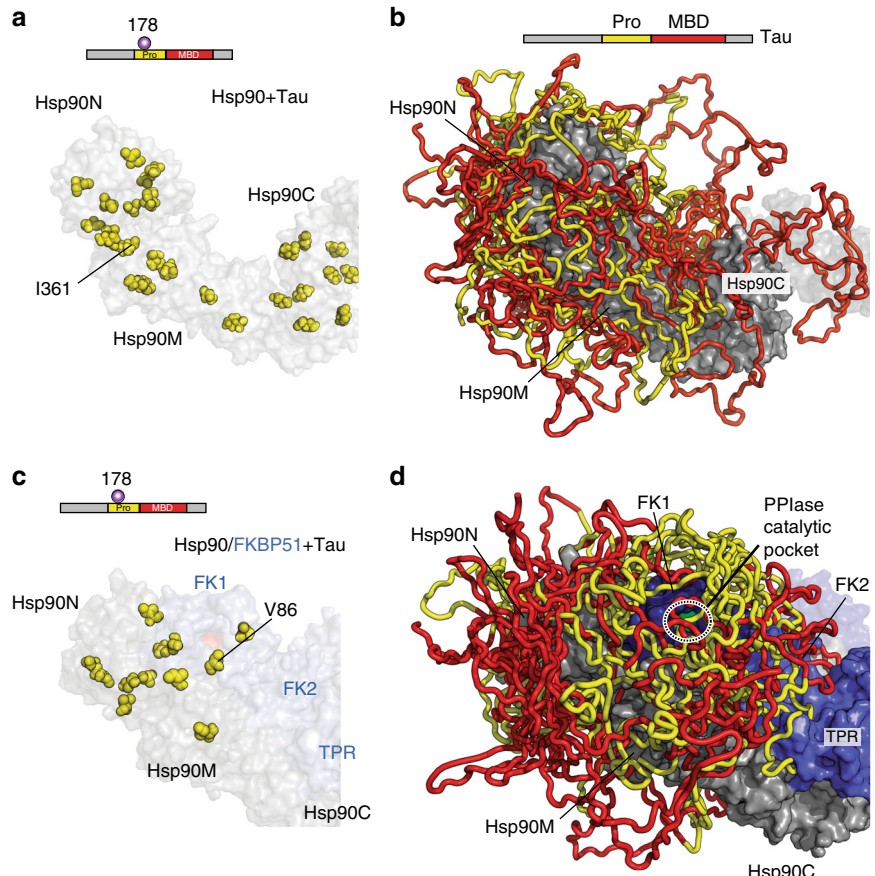

**Fig. 5** Tau's Pro-rich region targets to the catalytic site in the complex. **a**, **c** Isoleucine residues (yellow spheres) in Hsp90, which experience paramagnetic broadening by Tau tagged with MTSL at position 178 in the Hsp90/Tau complex (**a**) and the Hsp90/FKBP51/Tau complex (**c**). Hsp90 dimer is colored in gray, and FKBP51 in blue. Tau's domain organization is shown on top. **b**, **d** Representative ensembles of Tau conformations bound to Hsp90 (**b**) and the Hsp90/FKBP51 complex (**d**), which are in agreement with experimental intermolecular contacts. Twelve lowest-energy conformers are shown. Tau's proline-rich region (yellow) is targeted to the PPIase catalytic pocket of FKBP51 in the Hsp90/FKBP51/Tau complex. Tau's microtubule-binding domain is colored red. For clarity, Tau regions outside of the proline-rich domain and the MBD, which contribute little to the interaction, are not shown

**Promiscuity in chaperone/client interaction**. Tau is an intrinsically disordered protein, a class of proteins with an inherently high flexibility and adaptability[44,45]. To gain insight into the dynamic nature of the Tau/Hsp90 interaction, we compared paramagnetic effects induced in Hsp90 by Tau protein tagged with MTSL. Binding of Tau with MTSL at position 178 caused paramagnetic broadening of residues in Hsp90N. However, the same spin-labeled Tau also caused broadening of methyl groups in Hsp90M and even in Hsp90C, which is far away from Hsp90N (Fig. 5a, Supplementary Figure 7a, and Supplementary Table 5). Consistent with these data, Tau protein, which carried MTSL at position 352, induced broadening not only in Hsp90M and Hsp90C, but also in Hsp90N (Supplementary Figure 7b–c and Supplementary Table 5). The ability of Tau to bind to Hsp90 in multiple orientations was further supported by the distribution of XL-MS links between Tau and Hsp90 (Supplementary Figure 3b and Supplementary Table 3). The data show that binding of Tau to Hsp90 is highly polymorphic (Fig. 5b), in agreement with the inherent flexibility of Tau and other intrinsically disordered proteins[38,44–46], and is therefore best described by a broad ensemble of Tau conformations in the complex.

**Targeting Tau to the catalytic site in the complex**. To elucidate how the co-chaperone FKBP51 influences the interaction of Tau with Hsp90, we used Tau-induced paramagnetic broadening in Hsp90 and FKBP51. MTSL-mediated paramagnetic relaxation

enhancements (PREs) differed—despite the polymorphic nature of the interactions—between Hsp90/Tau and Hsp90/FKBP51/Tau (Fig. 5c, Supplementary Figure 7a–d, and Supplementary Table 5): paramagnetic broadening coming from Tau's proline-rich region was restricted to Hsp90N in the ternary Hsp90/FKBP51/Tau complex, but not in the Hsp90/Tau complex, that is, when the co-chaperone was not present (Fig. 5a–c). In addition, Tau's proline-rich region clusters around the catalytically active FK1 domain in the Hsp90/PPIase/client complex (Fig. 5c, d), in agreement with the distribution of cross-links observed between Tau and Hsp90/FKBP51 (Supplementary Figure 3b and Supplementary Table 4).

The data show that binding of the co-chaperone FKBP51 to Hsp90 introduces an orientational bias in the ensemble of bound Tau conformations. Contributions to this orientational bias could arise from the interaction of Tau's proline-rich region with the FK1 domain of FKBP51 (Supplementary Figure 3b). In addition, changes in the rotational freedom of Hsp90N[35]—through interaction with FKBP51 (Supplementary Figure 5)—might help in targeting Tau to the PPIase catalytic pocket.

## Discussion

Through a combination of structural and biochemical data, we gained detailed insight into the structural basis of the complex of unmodified, full-length human Hsp90 with the PPIase FKBP51. In addition, we studied the binding of the intrinsically disordered

protein Tau to this Hsp90/co-chaperone complex. Because of the complexity and dynamic nature of the interactions, a multidisciplinary approach based on complementary information from high-resolution NMR spectroscopy, XL coupled to MS, SAXS, size-exclusion chromatography (SEC), fluorescence and calorimetric binding studies, and activity measurements on full-length proteins and deletion constructs was essential to define and validate the structures of the 280-kDa dimeric and 340-kDa ternary complex.

Substrates and co-chaperones modulate the conformational landscape of Hsp90[5,25,47]. However, neither the co-chaperone FKBP51 nor the intrinsically disordered substrate Tau promoted allosteric changes towards more closed conformations (Fig. 1b and Supplementary Figure 1b). In addition, the interaction of Hsp90 with the PPIase FKBP51 was not affected by nucleotides, which favor more compact conformations. Importantly, even at excess of the PPIase FKBP51 the Hsp90/FKBP51 complex remained highly dynamic (Fig. 2i). The structure of the Hsp90/FKBP51 complex shown in Fig. 3 thus represents the final stable structure with fully bound FKBP51 and does not exclude intermediate states, in which only the TPR domain or the FK2 and TPR domain of FKBP51 are bound to Hsp90. Other co-chaperones, which bind elsewhere on Hsp90 (Fig. 6) and might promote closure of the dimer[5,7,25,48], could thus bind simultaneously with FKBP51 to Hsp90 and regulate the dynamic Hsp90/PPIase complex. Similar to the case of the TPR-containing protein Hop[49], the TPR domain of FKBP51 would remain tightly bound to the C-terminal dimerization domain of Hsp90 allowing for asymmetric ternary complexes[50] that would determine the conformation of substrates in complex with Hsp90[5].

The binding surface identified for FKBP51 on Hsp90 may be shared by all TPR-containing co-chaperones, which have been shown to compete for binding to Hsp90. Interestingly, FKBP51, unlike other co-chaperones, is significantly upregulated in the aged and AD brain[17]. This could cause an imbalance in Hsp90 heterocomplex dynamics as has been demonstrated by

displacement of the E3 ubiquitin-protein ligase CHIP, which targets substrates for proteasome degradation, from Hsp90 by FKBP51[17].

Intrinsically disordered proteins such as Tau are highly flexible and populate many different conformations in solution[51]. In addition, they can retain a significant amount of dynamics in complex with binding partners[38,44–46]. Our study shows that binding of Tau to Hsp90 and the Hsp90/FKBP51 complex is very dynamic, enabling different conformations of bound Tau and covering a vast region in Hsp90 and FKBP51 structures (Fig. 5). The mechanism of Tau/Hsp90 interaction is thus multivalent and pleomorphic, involving distant regions on Hsp90 structure. This mechanism of interaction is in contrast to a recently revealed chaperone/misfolded client complex, where a single conformation of the unfolded bound client was proposed[20]. The interaction of Tau with the Hsp90/FKBP51 complex thus expands the toolbox of chaperone/client recognition beyond single client-bound conformations[38]. This promiscuity could not only be due to Tau's inherent dynamics, but also related to Hsp90's client recognition mechanisms, since other Hsp90/client complexes have been shown to contain multiple binding modes in solution[40].

Our data also indicate that the primary role of Hsp90 in the ternary Hsp90/FKBP51/Tau complex is scaffolding, while FKBP51 provides co-chaperone action. Because Tau can directly bind to FKBP51 (Fig. 4d and Supplementary Figure 6)[17] and Tau and FKBP51 do not compete for the binding to Hsp90 (Fig. 6), Hsp90's role in the complex is to bring together Tau's proline-rich region and FKBP51's PPIase activity pocket (Fig. 5d). Moreover, the proline-rich region of Tau contains multiple Thr-Pro and Ser-Pro motifs, which are targets of proline-directed kinases and become phosphorylated in AD[16,52]. In addition, proline isomerization of Tau modulates Tau phosphorylation and oligomerization[53,54], with *cis*-proline, but not *trans*-proline, phosphorylated Tau appearing in the brains of humans with mild cognitive impairment[55]. Hsp90-based scaffolding might thus enhance Tau proline isomerization and Tau oligomerization, in agreement with the synergistic

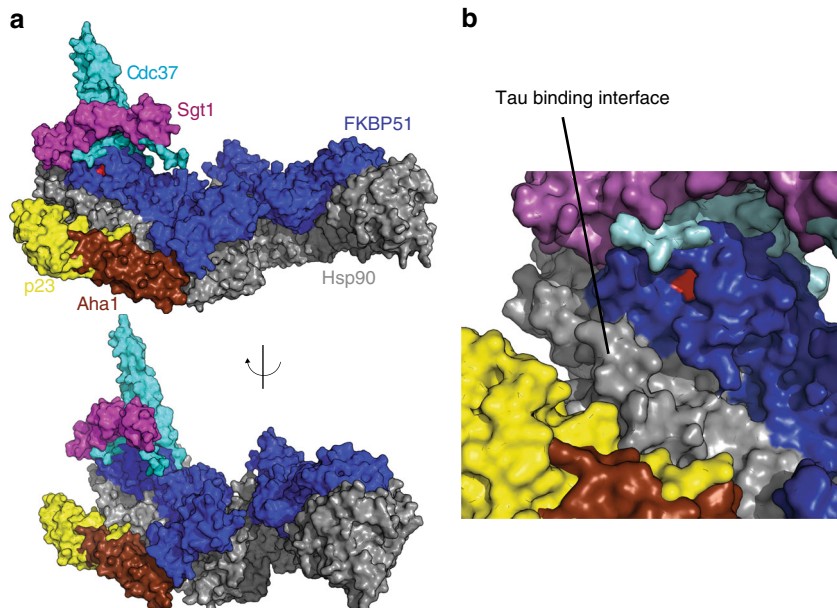

**Fig. 6** The ternary complex allows asymmetric Hsp90/co-chaperone interactions. **a**, **b** The Tau-binding interface on the Hsp90 surface is largely distinct from the binding sites of FKBP51 and other co-chaperones, which in turn do not overlap. PDB codes are: human Cdc37 (PDB id: 5fwk), *Arabidopsis thaliana* Sgt1 (PDB id: 2xcm), yeast p23 (PDB id: 2cg9), and yeast Aha1 (PDB id: 1usv). FKBP51's catalytic PPIase pocket, which is located in the FK1 domain, is highlighted in red

effect on Tau oligomerization by Hsp90/FKBP51 observed in vivo[17].

## Methods

**Sample preparation.** Human Hsp90β and human FKBP51 genes contained in a pET28b vector (Novagen) were used as templates for cloning different constructs: Hsp90NM (which lacks Hsp90C) as well as FK1–FK2 (which lacks TPR repeats) and FK2-TPR (which lacks FK1) from FKBP51. Boundaries for different domains are indicated in Figs. 1e and 2f. A TPR-only construct was also generated. However, the insolubility of the protein, even in a glutathione S-transferase-fused construct (through a pGEX5 vector, GE Healthcare), precluded its NMR assignment and use in fluorescence and calorimetric experiments. Oligonucleotides used for cloning are detailed in Supplementary Table 7. XL2-Blue Escherichia coli cells were used for cloning.

BL21 (DE3) and Rosetta 2 (DE3) E. coli strains were used for bacterial expression of all constructs. For uniform $^{13}C/^{15}N$ isotopic labeling, cells were grown in M9 minimal medium with $^{15}NH_4Cl$ and [D-$^{13}C$]-glucose as the sole sources of nitrogen and carbon, respectively. Metabolic precursors for selective $^1H$-$^{13}C$ labeling of Hsp90 isoleucine $\delta_1$ and FKBP51 leucine $\delta$, valine $\gamma$, alanine $\beta$, and methionine $\varepsilon$ methyl groups in fully deuterated media were purchased from NMR-Bio. Stereo-specific precursors were employed for labeling of FKBP51's leucine $\delta$ and valine $\gamma$ groups. Selective labeling was achieved as described in ref [56].

Cells were lysed by sonication and recombinant proteins purified by $Ni^{2+}$ affinity chromatography in Ni-NTA agarose (Thermo Fisher) using 50 mM Tris-HCl/500 mM NaCl/10 mM imidazole (pH 8.0) as binding buffer (elution buffer contained 250 mM imidazole). All Hsp90 variants were dialyzed and further purified through SEC using a buffer consisting of 10 mM HEPES/500 mM KCl/1 mM DTT (pH 7.5). FKBP51 variants were size-exclusion purified in 50 mM sodium phosphate/500 mM NaCl/10 mM DTT (pH 6.8). Superdex 75 and Superdex 200 columns (GE Healthcare) were used according to the proteins' molecular weights. To avoid possible interactions with the PPIase catalytic site in FK1 and to decrease the complexity of NMR spectra, His tags were removed from FKBP51 constructs through HRV3C protease cut (Thermo Fisher). Pure fractions were pooled, dialyzed into the desired buffer, concentrated, and stored at −80 °C until needed. Purified Hsp90 proteins were kept at concentrations ≤0.3 mM to prevent formation of higher-order oligomers. Possible protein aggregation was monitored by dynamic light scattering and only monomeric or dimeric (for full-length Hsp90) protein samples were used.

Expression and purification of htau40 was performed as described in refs. [57,58]. MTSL labeling of htau40 is described in ref [58]. In brief, full-length htau40 was expressed in the vector pNG2 in BL21 (DE3) E. coli strain. Pellets were resuspended in 50 mM MES/500 mM NaCl/1 mM MgCl₂/1 mM EGTA/5 mM DTT (pH 6.8) and cells were disrupted by French press and boiled during 20 min. Subsequently, recombinant protein was purified by FPLC SP-Sepharose column (GE Healthcare) using 20 mM MES/50 mM NaCl/1 mM MgCl₂/1 mM EGTA/2 mM DTT/0.1 mM PMSF (pH 6.8) as a binding buffer, increasing linearly the concentration of NaCl 1 M in the same buffer for elution. For MTSL labeling, DTT was removed from the samples by using SEC (PD-10 columns, GE Healthcare), and the proteins equilibrated in phosphate-buffered saline (PBS) buffer (pH 7.4). Fivefold molar excess of MTSL was added to attach MTSL to free sulfhydryl groups in ethyl acetate, at 21 °C for 2.5 h. Unreacted MTSL was removed by using PD-10 column and samples were concentrated and equilibrated in NMR buffer (with no DTT, see below).

Aha1 was cloned into pET21a vector (Novagen) and expressed in BL21 (DE3) cells. Cells were sonicated and purified with Ni-NTA agarose (Thermo Fisher) using 50 mM Tris-HCl/500 mM NaCl/10 mM imidazole (pH 8.0) as a binding buffer (elution buffer contained 250 mM imidazole). Then, after tobacco etch virus proteolysis, it was purified using SEC with 10 mM HEPES/500 mM KCl/1 mM DTT (pH 7.5) as buffer.

**NMR spectroscopy.** $^1H$-$^{13}C$ methyl-TROSY spectra[31] of selectively labeled Hsp90 and FKBP51 were acquired at 25 and 35 °C, respectively, on Bruker Avance III 800, 900, and 950 MHz spectrometers (all equipped with TCI cryoprobes) using 50 mM sodium phosphate/300 mM NaCl/1 mM DTT (pH 7.2) in 100% deuterium oxide ($D_2O$). To perform experiments in the presence of nucleotide (2 mM), spectra were acquired in 20 mM HEPES/5 mM KCl/10 mM MgCl₂/1 mM DTT (pH 7.4) in 100% $D_2O$. For NMR titrations, samples contained 80–100 µM selectively labeled protein and up to 5 molar equivalents of unlabeled partner. $^1H$-$^{15}N$ correlation spectra were acquired on $^{15}N$-labeled Tau containing 30–50 µM Tau and up to 5 equivalents of unlabeled partner in 50 mM sodium phosphate/10 mM NaCl/1 mM DTT (pH 6.8), and were performed at 5 °C to reduce amide proton-solvent exchange in Tau. At these conditions, Tau remains functional (i.e., can bind to microtubules) and monomeric[58], and based on the calculated binding affinities, the ternary complex is saturated (Fig. 4a–d). Upon binding to the different partners, the intensity of specific cross-peaks in 2D $^1H$-$^{15}N$ NMR correlation spectra of Tau decreased[58]. NMR binding plots show the inverse of the decay in signal intensity (Fig. 4d and Supplementary Figure 6b–c), where $I_0$ is the intensity of the cross-peaks of the reference spectra (Tau alone).

The presence of a paramagnetic center in Hsp90 will selectively decrease the intensity of FKBP51 residues, which are in spatial proximity (<25 Å) to the MTSL-carrying Hsp90 residue upon complex formation by inducing PRE[39,59]. Alternatively, the presence of the CLaNP-7 lanthanide tag in FKBP51 will induce PCSs on Hsp90 residues that are in close proximity (<25 Å) to the tag in the complex[59,60]. To label full-length Hsp90 with a paramagnetic center on a single site, the M362C mutation was generated (all native cysteine residues of human Hsp90 are buried and thus not available for tagging). Tagging of unlabeled Hsp90$_{M362C}$ with MTSL (using both paramagnetic and diamagnetic MTSL) was performed as described in ref [58]. $^1H$-$^{13}C$ methyl-TROSY spectra[31] experiments were used at 35 °C to observe PRE broadening induced by Hsp90$_{M362C}$-MTSL onto Leu and Val cross-peaks of selectively Leu/Val-labeled FKBP51 (using 60 µM FKBP51$_{LeuVal}$ and 30 µM Hsp90$_{M362C}$-MTSL). In order to tag full-length FKBP51 with the CLaNP-7 lanthanide tag[60] on a single site, the E110C mutation was introduced. To avoid unspecific tagging on the structure of FKBP51, the rest of the cysteine residues were removed through the mutations C176S, C183S, C215S, C339S, and C394S. Unlabeled, full-length FKBP51$_{C176S/C183S/C215S/C339S/C394S/E110C}$ was purified as the wild-type protein and its proper fold was confirmed by circular dichroism spectroscopy. Subsequently, CLaNP-7 was attached to the C107/E110C residues in FK1, using both Tm$^{+3}$-attached and Lu$^{+3}$-attached (paramagnetic and diamagnetic lanthanides, respectively) clamps as described in ref [60]. $^1H$-$^{13}C$ methyl-TROSY spectra[31] experiments were acquired at 25 °C to observe PCSs induced by CLaNP-7-tagged FKBP51$_{C176S/C183S/C215S/C339S/C394S/E110C}$ onto Ile cross-peaks of selectively Ile-labeled Hsp90 (using 80 µM Hsp90 and 40 µM FKBP51$_{C176S/C183S/C215S/C339S/C394S/E110C}$-CLaNP-7). Because the formation of the complex broadens both Hsp90 and FKBP51 methyl resonances (Figs. 1d and 2a–d), PRE and PCS experiments were performed in sub-saturation conditions to enable detection of PRE/PCS effects with sufficient sensitivity. PREs were extracted from the ratio of peak intensities of the paramagnetic and diamagnetic samples (for Hsp90$_{M362C}$-MTSL). PCSs are the chemical shift changes of Hsp90 cross-peaks between Tm$^{+3}$-loaded and Lu$^{+3}$-loaded samples (for FKBP51$_{C176S/C183S/C215S/C339S/C394S/E110C}$-CLaNP-7). Experiments were performed in 50 mM sodium phosphate/300 mM NaCl (pH 7.2) in 100% $D_2O$. Proper tagging of Hsp90$_{M362C}$ and FKBP51$_{C176S/C183S/C215S/C339S/C394S/E110C}$ with the different spin labels was confirmed by absorbance at 590 nm[60] and 1D NMR. Orientation of the lanthanide tensor (Fig. 3f) was obtained as described in refs [60,61]. The z-axis of the tensor points in the direction of the Cys107 and Cys110 side chains, with the x-axis and y-axis making angles of 20° and 5° with the Cα-Cα vector of the two cysteine residues, respectively[60,61].

For characterization of the interaction of Tau with Hsp90 and Hsp90/FKBP51, 30 µM of MTSL-labeled Tau (to avoid undesired solvent PREs[62]) were mixed with up to two equivalents of methyl selectively labeled Hsp90 and Hsp90/FKBP51 at 25 °C in 50 mM sodium phosphate/300 mM NaCl (pH 7.2) in 100% $D_2O$. PREs were extracted from the ratio of peak intensity of the paramagnetic sample and the peak intensity from the same sample after incubation with 10 mM DTT, which cleaves off the paramagnetic tag from Tau (diamagnetic state).

3D $^{13}C$-[$^1H$-$^1H$]-NOESY and 4D $^1H$-[$^{13}C$-$^{13}C$]-$^1H$-NOESY experiments (both with 200 ms mixing time) were acquired on samples containing 300 µM of methyl-labeled full-length Hsp90, FKBP51, and equimolar (1:1) mixtures of both proteins at 35 °C on Bruker Avance III 950, 900, and 800 MHz spectrometers (all equipped with TCI cryoprobes), using the buffer 50 mM sodium phosphate/300 mM NaCl/1 mM DTT (pH 7.2) in 100% $D_2O$. 4D NOESY experiments[34] were acquired using non-uniform sampling (with 3% sampling density) and reconstructed using the SMILE algorithm[63].

Human Hsp90β contains 48 isoleucine residues well spread over all domains (Supplementary Figure 2a), including Hsp90N, the charged flexible linker (CL), which connects Hsp90N and the middle domain (Hsp90M)[37], and Hsp90C, responsible for dimerization. FKBP51 contains two domains (termed FK1 and FK2) and three flexible TPR repeats (PDB id: 1kt0, 5njx)[64,65]. Previously available assignments of Hsp90 isoleucine methyl groups were confirmed and extended through site-directed mutagenesis of specific isoleucine residues in both full-length Hsp90 and isolated N–M, N, and M domains[66]. In addition, 3D HNCO, 3D HNCACB, 3D HNCOCA, 3D HcCH-TOCSY (15 ms mixing time), 3D $^{15}N$-[$^1H$-$^1H$]-NOESY (80 ms mixing time), and 3D $^{13}C$-[$^1H$-$^1H$]-NOESY (80 ms mixing time, both aliphatic and aromatic) NMR experiments were employed at 35 °C on 300 µM of $^{15}N/^{13}C$ and $^2H/^{15}N/^{13}C$ uniformly labeled Hsp90M samples, and on 400 µM of $^{15}N/^{13}C$ uniformly labeled FK1, FK2, and FK1–FK2 domain samples for full assignments. Assignments obtained from individual domains were compared to spectra of the full-length proteins, indicating that all individual domain constructions used in this work remained properly folded. Spectra were processed using nmrPipe[67] and TopSpin, and analyzed in Sparky[68] and ccpnmr Analysis 2.2.1. While FK1 and FK2 domains from FKBP51 were fully assigned (the TPR region remained unassigned, Supplementary Figure 2b), 44 out of 48 isoleucine residues present in full-length Hsp90 (38 out of 38 excluding the C-terminal domain) were sequence specifically assigned.

**Chemical XL and MS.** For XL experiments, 125 pmol of each complex were cross-linked with a 50× molar excess of freshly prepared BS3 (bis(sulfosuccinimidyl) suberate, Pierce Biotechnology) in 10 mM sodium phosphate/100 mM NaCl/2 mM MgCl₂ (pH 7.4) for 30 min at room temperature with gentle mixing. The reaction was quenched by addition of 5 µl of 2 M Tris-HCl buffer (pH 8.0). The complexes

were precipitated by adding fourfold volume of pre-chilled acetone and incubated at −20 °C for more than 2 h. The precipitate was spun down, washed with pre-chilled 70% ethanol, and re-dissolved in 50 mM NH₄HCO₃/4 M urea (Sigma-Aldrich) (pH 8.0). After reduction and alkylation, the urea concentration was taken to 1 M and proteins were digested with trypsin (Promega) 1:20 w/w ratio overnight at 37 °C.

Tryptic peptides were desalted using Sep-Pak SPE C18 cartridges (Waters) and separated using SEC[69,70]. Peptides were separated on a nano-liquid chromatography system (UltiMate™ 3000 RSLCnano system) and analyzed on an LTQ-Orbitrap Velos mass spectrometer (Thermo Fisher) according to ref. [71]. Data analysis was performed using pLink as described[72] using a target-decoy strategy. Database search parameters included 10 ppm tolerance in the survey scan, 20 ppm tolerance in the product ion scan, carbamidomethylation on cysteines as a fixed modification, and oxidation on methionines as a variable modification. The number of residues of each peptide on a cross-link pair was set between 4 and 40. A maximal of two trypsin missed-cleavage sites were allowed. The results were obtained with 1% false discovery rate. Cross-links were visualized with xiNET (http://crosslinkviewer.org). Experiments were performed up to six times and non-reproducible cross-links were discarded.

**SAXS**. SAXS data were collected at 25 °C from pure and monodisperse samples of Hsp90, FKBP51, Tau, and the different mixtures in 50 mM sodium phosphate/10 mM NaCl/1 mM DTT (pH 6.8). The 20 mM HEPES/5 mM KCl/10 mM MgCl₂/1 mM DTT (pH 7.4) buffer was used to test the effect of nucleotide binding (2 mM) on the shape of the complexes. When necessary, protein complexes were formed in the presence of nucleotide prior to the measurements. Sample concentrations ranged from 20 to 100 μM, and shown $P(r)$ distributions contained 50 μM Hsp90 and the corresponding equivalents of Tau and FKBP51 (Fig. 1b). Given the binding constants of Hsp90/FKBP51, Hsp90/Tau, and Hsp90/FKBP51/Tau (Figs 2i and 4a), the ternary complex was formed in the chosen experimental conditions (Fig. 1b). Therefore, a mixture of binary complexes can be ruled out. Scattering profiles were analyzed using the ATSAS package and standard procedures. Reported values were calculated merging the curves obtained from replicates at different concentrations. SAXS measurements were performed at DESY (Hamburg, Germany) and Diamond Light Source (Oxford, UK) stations.

**SEC-MALS**. Samples subjected to SEC coupled to multi-angle light scattering (SEC-MALS) contained 15 μM Hsp90/60 μM FKBP51/90 μM Tau (containing either full-length Tau or K32 fragment). Complexes were incubated overnight at room temperature, followed by 1 h incubation at 37 °C, prior to injection into the SEC-MALS (150 μl) in 25 mM NaP/300 mM NaCl (pH 7.2) buffer (samples contained additionally 1 mM DTT). For analysis of the complexes (Fig. 4b, c), a Superose 6 10/300 GL column (GE Healthcare) was connected to an Agilent 1260 Infinity HPLC system, miniDAWN TREOS multi-angle light-scattering device (Wyatt Technology), and an Optilab T-rEX refractive index detector (Wyatt Technology). A Superdex 200 10/300 GL column (GE Healthcare) was used for the analysis of isolated proteins (Supplementary Figure 1a). Data collection and analysis were performed using the ASTRA 6.0.2 software (Wyatt Technology).

**Fluorescence-binding experiments**. IANBD amide (N,N′-dimethyl-N-(iodoacetyl)-N′-(7-nitrobenz-2-oxa-1,3-diazol-4-yl)ethylenediamine; Thermo Fisher Scientific) fluorescent dye was used for specific labeling of the Hsp90ₘ₃₆₂ₓ mutant, which contains a single solvent accessible Cys in each protomer. Proteins were incubated with a fivefold molar excess of the dye while gently shaking overnight at 4 °C. Unreacted dye was removed from the labeled protein with a PD-10 desalting column (GE Healthcare) using 25 mM sodium phosphate/300 mM NaCl (pH 7.2) buffer. Kinetic experiments were carried out with an Applied Photophysics SX.20 stopped-flow spectrophotometer at 37 °C using 25 mM sodium phosphate/300 mM NaCl at pH 7.2 as buffer. IANBD-labeled Hsp90ₘ₃₆₂ₓ (1 μM) was mixed with equal volumes of FKBP51, FK1–FK2, FK2-TPR, and buffer (as control), under pseudo-first-order conditions. The excitation wavelength was set to 465 nm and a 495 nm cutoff filter was used to collect IANBD emission ($n$ = 3–5 technical replicates for each tested protein). Due to the intrinsic fluorescence increase of Hsp90 in buffer (probably due to dimer dissociation upon dilution and fluorescence self-quenching in the dimer), the rates of the interaction were calculated as fluorescence increase at 20 s (when the control experiments reached the plateau). For affinity calculations, IANBD fluorescence emission was measured at 37 °C with a Fluorolog-3 spectrophotometer (Model FL322, HORIBA Jobin Yvon) in the range of 500 to 620 nm with an excitation wavelength of 465 nm and 5-nm slit width using 0.5 μM Hsp90, or Hsp90/FKBP51 (1:1), and increasing concentrations of Tau. Both Hsp90 alone and Hsp90/FKBP51 complex were incubated for 2 h at 37 °C before the experiment. Reported are average values of triplicate experiments ± SD.

For FKBP51/Tau affinity determination, fluorescence spectra were acquired on an Agilent Cary Eclipse Fluorescence Spectrophotometer (excitation wavelength of 295 nm, emission 300–400 nm) at 25 °C in 25 mM HEPES/5 mM MgCl₂/10 mM KCl/1 mM DTT (pH 7.2) buffer in a microcuvette with 10 mm path length. Before measuring the excitation profile, 10 μM FKBP51 were incubated for 5 min after addition of Tau. The concentration of FKBP51 was kept constant throughout the experiment. Data were fitted using nonlinear least-square fitting, according to the

equation $F = F_{min} + A* [Tau]^n/(K^n + [Tau]^n)$, where $F$ is the percentage of fluorescence quenching, $F_{min}$ is the minimal fluorescence quenching, $A$ is the difference between maximal and minimal fluorescence quenching, and $K$ is the dissociation constant. The error was calculated for each Tau concentration using standard deviation of three independent experiments.

**Isothermal titration calorimetry**. Isothermal titration calorimetry (ITC) was carried out with a Microcal PEAQ-ITC automated (Malvern) at a constant temperature (10 °C) in 25 mM HEPES/10 mM KCl/5 mM MgCl₂/1 mM DTT (pH 7.2). AMP.PNP (adenylyl-imidodiphosphate, 2 mM) was included in the buffer to determine the effect of nucleotides in binding affinities. Deionized water was used in the reference cell. All solutions were thoroughly degassed before use. We determined the reproducibility of the measurements by using replicates of the independent measurements. The sample cell contained Hsp90 at 20 μM and was titrated with 200 μM of FKBP51. For the titration of Hsp90 with FK2-TPR, we used 25 μM of Hsp90 in the sample cell and titrated it with 140 μM of FK2-TPR. At each titration step, a 0.4 μl injection was followed by 18 injections of 2 μl into the sample cell, with a stirring speed of 750 rpm and intervals of 150 s between injections. The data were fitted using the Microcal PEAQ-ITC Analysis software. We obtained heats due to dilution by applying a fitted offset parameter in the analysis software to subtract a constant heat from all the injection points. Thermodynamic parameters were obtained by fitting a macroscopic binding model allowing for one set of binding sites. Changes in stoichiometry of the interaction were attributed to partial inactivation of Hsp90 in the experimental conditions.

**Structural modeling**. Structural modeling was performed with the program Haddock[73] using both full-length human Hsp90β (PDB id: 5fwk)[19] and FKBP51 (PDB id: 1kt0, 5njx)[64,65] as template structures. For the final refinement steps, an additional atomic structure encompassing FK1 domain with an additional α-helix in the N-terminus (PDB id: 3o5e) was used. To drive the calculations towards Hsp90/FKBP51 complex formation, we used the observed intermolecular NOEs (Supplementary Table 1). Intermolecular NOEs were defined as unambiguous distance restraints with a cutoff of 6 Å between the specific methyl groups[74]. In addition, ambiguous distance restraints (as defined by Haddock) of 30 Å[75] between the Cα atoms of residues showing intermolecular cross-links were enforced (Supplementary Table 2). The extended Hsp90 dimer was used as a platform (PDB id: 5fwk) and was modified by rotation of the Hsp90M/C interface as observed in ref. [76] for Hsp90 in the apo form, in order to fit the SAXS shape envelope (Fig. 1b).

Due to extensive interdomain dynamics in Hsp90[35], the final structure was constructed using the different domain pairs as building blocks. First, a structure including Hsp90M-C-C-M of the dimer (therefore excluding the loosely coupled Hsp90N domains)[35] was docked with FK1–FK2 from FKBP51. The FK1–FK2 interdomain region remains rigid in solution as concluded from the observed interdomain NOEs and additional residual dipolar coupling measurements, which showed that the relative domain orientation of the FK1–FK2 domains as observed in the crystal structure of FKBP51 (PDB id: 1kt0 and 5njx) is present in solution. The best 20 docking results were inspected according to the experimental observations and the final structure refined, and used as a template for consecutive steps. Next, Hsp90N was added to the Hsp90MCCM/FK1–FK2 complex, and after structure inspection and refinement, used as a template for the final addition of the TPR repeats onto the full Hsp90/FK1–FK2 complex. Only a minimal rotation of the flexible TPR repeats was needed, in order to accommodate both the MEEVD-binding site[65] and Hsp90C's flexible loop[76–78] in the model of the complex. The structure of the complex was validated by experimental PCSs and PREs obtained by tagging both partners of the complex (Fig. 3d–g). Docking statistics for the refined structural model are shown in Supplementary Table 6.

To determine representative ensembles of Tau conformations bound to Hsp90 and Hsp90/FKBP51, which are best in agreement with the experimental intermolecular NOEs and XL-MS cross-links, we used as template the structural ensemble of Tau in solution as derived from experimental NMR and SAXS data[79]. Since Tau binding does not promote conformational changes on Hsp90 dimer (Supplementary Figure 1b), the extended Hsp90 dimer structure, which had been built for the Hsp90/FKBP51 complex, was used as template to construct the Hsp90/Tau complex (Fig. 5b). The Hsp90/FKBP51 structure was used to build the Hsp90/FKBP51/Tau complex (Fig. 5d). Because only Tau's proline-rich and microtubule-binding regions bind to Hsp90 and Hsp90/FKBP51 (Fig. 4d), those were the only Tau regions included in the docking calculations. To this end, ten random Tau conformers were selected from the ensemble of unbound Tau. To take into account the dynamic nature of the Tau molecule, structures of all Tau conformers, which were defined as fully flexible in Haddock, were divided into individual parts covering the region of interest: residues 151–198, 199–244, 245–310, and 311–369 from 2N4R Tau[16]. Intermolecular cross-links were enforced as experimental ambiguous restraints with 30 Å cutoff[75] between the Cα atoms of the cross-linked residues. In addition, PREs were implemented as ambiguous restraints to the specific Hsp90 and FKBP51 methyl groups. No PRE-based restraints were used for the Tau/TPR interaction due to a lack of sequence-specific assignments of the TPR region. Because the strength of PRE broadening in dynamic systems such as Hsp90/Tau and Hsp90/FKBP51/Tau is influenced by several factors[39,62], we defined ambiguous distance restrains of 15, 20, and 25 Å (according to the strength of PRE-induced broadening of individual methyl cross-peaks) in Haddock docking

calculations. In case of the Hsp90/Tau complex, all Tau fragments derived from the ten Tau conformations of unbound Tau were docked to the individual Hsp90 domains. Hundreds of bona fide ensembles containing Tau fragments in complex with Hsp90 were generated and only those where the free connecting ends of Tau fragments were in spatial proximity (no spatial restrains were introduced between the connecting ends of the different Tau fragments in the docking calculations) were manually connected to build the full conformers of Hsp90-bound Tau. Resulting complex structures were manually curated and those fulfilling the experimental restraints were used to generate the final representative ensemble of Tau bound to Hsp90. A similar strategy was followed to generate a representative ensemble of Tau conformations bound to the Hsp90/FKBP51 complex. In this case, all Tau fragments were selected from ten conformers of free Tau and docked to Hsp90N, Hsp90M/FK1, Hsp90M/FK2, and Hsp90CC/TPR. The resulting ensembles were inspected, refined and annealed to the final ensemble. Complex asymmetry was not imposed in our calculations since it cannot be derived from our experimental observations; however, the structural models shown in Fig. 5 are asymmetric based on a Tau/Hsp90 complex previously described[27]. Because of the high flexibility of intrinsically disordered proteins, the obtained ensembles of Tau conformations are representative for the experimental data, but do not exclude additional conformations/ensembles. Structures were displayed using PyMOL[80].

**ATPase activity assay.** The P$_i$Per$^{TM}$ Phosphate Assay Kit from Molecular Probes (Invitrogen) was used to measure the ATPase activity of Hsp90. Prior to measurements, protein samples were dialyzed into 20 mM HEPES/10 mM KCl/5 mM MgCl$_2$/1 mM DTT (pH 7.4). Samples contained Hsp90 at 20 μM and other proteins in stoichiometric ratios as required. ATP (2 mM) was used as the substrate in the reaction. A TECAN GENios Pro plate-reader was utilized at 37 °C over a course of 2 h to measure 121 individual absorbance data points at 562 nm. Absorbance data were plotted against time to yield a curve corresponding to the ATP hydrolyzed over the course of reaction. The maximum slope was then taken as the rate of ATP hydrolysis. All measurements were performed in triplicate and reported values include the standard error of the mean.

## Data availability

Additional relevant data are available from the corresponding author upon reasonable request.

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

## Acknowledgements

We thank S. Hübschmann (DZNE Bonn) and S. Cima (DZNE Göttingen) for preparation of Tau samples, J. Ying (NIH, MD) for help in reconstruction of 4D NOESY experiments, U. Plessman (MPIBPC, Göttingen) for assistance in MS, and S. Ambadipudi (DZNE Göttingen) for MARK2 phosphorylation of Tau. We are grateful to M. Ubbink (Leiden University, The Netherlands) for the CLaNP-7 lanthanide tag. This work was supported by the National Institutes of Mental Health Grant R01 MH103848. J.O. was supported by a Marie Curie Intra-European fellowship (project number 626526) and B.J.C. by a Fulbright scholarship. M. Z. was supported by the European Community's Seventh Framework Programme (FP7/2007-2013) under BioStruct-X (grant agreement 283570), and the advanced grant '87679 – LLPS-NMR' of the European Research Council. H.U. and M.Z. were supported by the German Science Foundation (Collaborative Research Center 860; projects A10 and B2). L.J. B. was supported by the NIH/NINDS R01 NS073899 Grant.

## Author contributions

J.O. designed the project, conducted protein preparation, SAXS, XL, and NMR data acquisition and analysis and structural modeling by molecular docking; B.J.C. performed protein preparation and NMR analysis for the assignment of Hsp90 residues and ATPase measurements; P.W. performed NMR analysis for the assignment of FKBP51 residues; P. C. performed ITC and fluorescence measurements; A.P.-L. performed fluorescence experiments and analysis; C.-T.L. and R.V.H performed XL-MS experiments and analysis; J.B. and J.D.B. prepared samples for NMR; M.Z., H.U., E.M., C.A.D., and L.J.B. designed and supervised the project; J.O. and M.Z. wrote the paper.

## Additional information

**Competing interests:** The authors declare no competing interests.

