## [Peer Review File · Nature Communications]

Reviewers' comments:

Reviewer #3 (Remarks to the Author):

This manuscript studies the interaction of chaperone Hsp90, co-chaperone FKBP51 and the disordered client Tau. By combining multiple methods, the authors firstly show that the N and M domains of Hsp90 and the FK1 and FK2 domains of FKBP51 are the ones involved in the interaction between these proteins. A structural model of Hsp90/FKBP51 is built based on several experimental approaches and further validated by PRE and PCS. To support their result, they also add some quantitative affinity data and dynamic binding result of Hsp90/full length FKBP51, Hsp90/FK1 deleted FKBP51. I think the PCS (Fig. 3f-g) and ITC data (Supplementary figure 5b-c) the authors added is solid evidence to support this part of conclusion. Although further atomic level result are lacking, considering the large size of proteins and large continuous binding area, the structural information provided by the authors provides good progress towards understanding of the Hsp90 and co-chaperone interaction mechanism. I have some questions below about the calculation (question 2) and detail of Hsp90/FKBP51 interaction (question 3) for the authors to clarify or answer.

For the Tau/Hsp90/FKBP51 ternary complex, authors also provided more evidence from different techniques, including more NMR titration, SEC and SAXS, to support the formation of ternary complex. But I still have some doubts about it (see question 1).

Question 1:

About ternary complex, or

about the platforming role of Hsp90 or the biological significance of the ternary complex.

Reviewer 1 questioned the "platforming" role of Hsp90. The authors reply in their rebuttal (see the rebuttal file, page 1-2), but I think there are still some problems here for the ternary complex:

1. Authors say FKBP51 does not compete with Tau for binding Hsp90, but instead is "platforming". However, there is no evidence that Hsp90 changes the affinity of FKBP51 and Tau. We could just say that Hsp90 binds two different proteins with no interaction between them. If Hsp90 acts as a "platform", I would expect it to show some changes in the presence of the two proteins.
2. In the rebuttal, page 2 paragraph 1, authors say "(Fig. 4a) shows that Hsp90's binding affinity for Tau slightly increases in the presence of FKBP51". But actually, Fig. 4a shows that the K_d value of Hsp90/Tau ($4.2 \pm 1.3 \mu\text{M}$) is lower than (Hsp90+FKBP51)/Tau ($8.2 \pm 0.7 \mu\text{M}$), suggesting higher affinity for Tau in the absence of FKBP51.
3. In the new figure/Data 2, the authors say "In addition, we also performed NMR titration experiments adding increasing equivalents of FKBP51 to a preformed Tau:Hsp90 (1:2) complex (see new figure 2). There is almost no change in Tau's binding profile in the presence of increasing amounts of FKBP51, suggesting that Tau binds to Hsp90 more strongly than to FKBP51, which, at our experimental conditions, is bound to Hsp90 (based on the nanomolar binding affinity for the Hsp90:FKBP51 complex, Supplementary Fig. 5b)". But the problem is, FKBP51 itself would decrease the intensity of Tau too! So the "no change in Tau's binding profile" could be also caused by Tau bind to FKBP51, which compensates for the change caused by Hsp90. To support the authors' conclusion, they need to find there are some specific residues of Tau that only change in Hsp90 binding or only in

FKBP51 binding. Otherwise this supporting result is not very convincing to conclude that Tau preferably bind Hsp90 but not FKBP51 in the ternary complex. Similarly, I think Figure 4d has same problem, in which the authors suggest that “Chemical shift changes induced in Tau by the Hsp90/FKBP51 complex were however very similar to those observed with Hsp90 alone (Fig. 4d)”. But again, authors could not rule out that FKBP51 itself could affect Tau too, meaning that in the sample of Tau+Hsp90+FKBP51 (1:2:2) in Fig. 4d, some portion of Tau bind FKBP51 only and could also contribute to signal attenuation. Besides, if Tau bind to the Hsp90 in Hsp90/FKBP51 complex, we should expect the intensity change be larger than the Tau/Hsp90, because Hsp90/FKBP51 is a larger complex than Hsp90 alone, which will cause more signal reduction. But from Fig. 4d, Tau+Hsp90+FKBP51(1:2:2) is similar or even a little weaker than Tau+Hsp90(1:2).

4. Authors say “chromatographic measurements demonstrated that FKBP51 stabilizes the ternary complex (Fig. 4b-c)” (line 215). But to me, the asterisk peaks, which are ternary complex suggested by the authors, contain mostly Hsp90 and FKBP51 but not Tau or K32 (I can't find which band is Tau, is it completely overlapped with FKBP51?). I wonder what the chromatograph of Hsp90-FKBP51 looks like (I am surprised the authors did not show it. In the New Figure/Data 3 they show K32 and FKBP51, but I did not find any SEC of Hsp90 and FKBP51 complex). It could help to tell that the three samples chromatography are different from the 2 samples' one, suggesting the existence of ternary complex.

5. In my opinion, the most significance of the ternary complex, is the existence of FKBP51 would introduce an orientational bias in the ensemble of Tau conformations in Hsp90 binding (“Targeting Tau to the catalytic center in the Hsp90/PPIase complex” in Results, line 278-296). This suggest how co-chaperone might remodel the Hsp90's client binding. But this conclusion is only supported by one PRE experiment (178 of Tau). As they also labeled at 352 of Tau for testing interaction with Hsp90 (Supplementary Figure 6b), I wonder how this would look like when adding Hsp90/FKBP51 complex. Besides, authors say “In addition, Tau's proline-rich region clusters around the catalytically active FK1 domain in the Hsp90/PPIase/client complex (Fig. 5d)”. I did not understand how the PRE result supports this. Moreover, does it contradict the authors' other conclusion that Tau preferentially binds Hsp90 only in the ternary complex?

Question 2:

About structural model calculation and validation

I do not agree with the reviewer 2's comment that “The Hsp90-FKBP51 complex was built on the basis of just (!) 7 inter-molecular NOEs. The interacting surface is very large and typically would require several hundred of NOEs to be reasonably defined.” I think when considering inter-molecular NOEs, the reliability is more important than the number. A few reliable inter-molecular NOEs combine with some other ambiguous restraints could be applied to get a very reasonable structural model. As it is a structural model, we could not ask for high accuracy at every residue. Rather, the further experimental validation would be important for testing the model and provide a solid conclusion.

There appears to be some problem or lack of clarity of the data in the model building:

1. In the Results (line 175-187) and Methods (line 609-641), the authors say they use unambiguous inter-molecular NOEs and ambiguous restraints from cross-links to calculate the model, and use PRE and PCS to validate the model. I think it is more reliable to add the PRE and PCS in the model calculation (at least final model calculation), especially the very decent PCS result. I think it would be very helpful for the convergence and clustering. I wonder how the model would look if it was calculated with just the inter-molecular NOEs, PRE and PCS.

2. In the rebuttal, authors say the Hsp90-FKBP51 complex calculation are using data from following parts:

- Residue-specific NMR chemical shift perturbation
- PCSs and PRE data
- 16 inter-molecular crosslinks
- 7 inter-molecular NOEs
- SAXS data

But I have some questions here: 1) about the “Residue-specific NMR chemical shift perturbation”, I did not see this mentioned elsewhere in the paper, either in Results or Discussion. What does the CSP look like? 2) the authors mention here that they use PCSs and PRE data for calculation, but in the result and method part, they mention that PRE and PCS are used to validate the model. Which one is true? They cannot use PRE and PCSs for both calculation and validation. The common strategy is use the PRE and PCSs for model calculation. For validation, mutagenesis and truncation provide more solid result. The authors need to clarify the model structural calculation method.

Question 3:

About some detail of Hsp90-FKBP51 interaction

1. Since the authors could detect inter-molecular NOEs between Hsp90N/FK1, Hsp90CL/FK1, Hsp90M/FK1, Hsp90M/FK2, Hsp90M/FK2, at equimolar protein, it is puzzling that, in their titration experiment, FK1 and FK2 do not show significant signal attenuation even with 2 fold addition of Hsp90, but requires a 4 times excess of Hsp90. Could this attenuation cause by the nonspecific binding of FK1 and FK2 to more than one Hsp90 under these conditions of excess Hsp90? It is rather dubious to suggest the presence of specific cross-links, PREs and NOEs, for a weak nonspecific interaction.
2. In Fig. 2h, concerning the interaction of Hsp90 and FK2TPR, deleting FK1 domain mildly decreases the Hsp90 binding affinity (Supplementary figure 5) and weakly accelerates the binding rate (Fig. 2h). Is this because FK1 is present in bound and unbound conformations? Could it also be caused by the rotation of Hsp90N to accommodate FK1 in the complex as suggested by the authors (line 193-198)? About the rotation of Hsp90N in the complex structure, how reliable is the evidence of this? According to the Methods (line 623-641), the authors first calculate the Hsp90MCCM/FK1FK2 complex, and then Hsp90N is added after that. Could the observation be an artifact of the conditions of the calculation?

Some minor questions about analyzing the intensity profile.

1. For some NMR titrations, authors use very high concentration ratio. In Fig. 2e, the FK1 and FK2 domain of FKBP51 show intensity change after adding 4-5-fold of Hsp90. Considering the relative high concentration of sample they use (~100uM of labeled FKBP51), these change may be caused by unspecific binding and thus is not biological relevance.
2. In Fig. 4d and new Figure/Data 1, by comparing the intensity change of Tau with Hsp90 and FKBP51, the authors conclude that as the change of Tau with Hsp90 is more significant, Tau binding Hsp90 is stronger than with FKBP51. I think it is not very convincing because the difference of the signal attenuation in Tau could be caused by the different size and dynamics of Hsp90 and FKBP51. To carefully make this conclusion, authors should measure the Kd value of Tau and FKBP51, as reviewer 1 suggests.

Reviewer #3: This manuscript studies the interaction of chaperone Hsp90, co-chaperone FKBP51 and the disordered client Tau. By combining multiple methods, the authors firstly show that the N and M domains of Hsp90 and the FK1 and FK2 domains of FKBP51 are the ones involved in the interaction between these proteins. A structural model of Hsp90/FKBP51 is built based on several experimental approaches and further validated by PRE and PCS. To support their result, they also add some quantitative affinity data and dynamic binding result of Hsp90/full length FKBP51, Hsp90/FK1 deleted FKBP51. I think the PCS (Fig. 3f-g) and ITC data (Supplementary figure 5b-c) the authors added is solid evidence to support this part of conclusion. Although further atomic level result are lacking, considering the large size of proteins and large continuous binding area, the structural information provided by the authors provides good progress towards understanding of the Hsp90 and co-chaperone interaction mechanism.

Reply: We thank the reviewer for her/his supportive words.

Reviewer #3: I have some questions below about the calculation (question 2) and detail of Hsp90/FKBP51 interaction (question 3) for the authors to clarify or answer. For the Tau/Hsp90/FKBP51 ternary complex, authors also provided more evidence from different techniques, including more NMR titration, SEC and SAXS, to support the formation of ternary complex. But I still have some doubts about it (see question 1). Question 1: About ternary complex, or about the platforming role of Hsp90 or the biological significance of the ternary complex. Reviewer 1 questioned the “platforming” role of Hsp90. The authors reply in their rebuttal (see the rebuttal file, page 1-2), but I think there are still some problems here for the ternary complex: 1. Authors say FKBP51 does not compete with Tau for binding Hsp90, but instead is “platforming”. However, there is no evidence that Hsp90 changes the affinity of FKBP51 and Tau. We could just say that Hsp90 binds two different proteins with no interaction between them. If Hsp90 acts as a “platform”, I would expect it to show some changes in the presence of the two proteins.

Reply: We thank the reviewer for raising this point. The platforming role, which is attributed to Hsp90 in the ternary complex, is based on the different affinities of Hsp90 to FKBP51 ($K_d = 249$ nM, Supplementary Figure 5b) and to Tau ($K_d = 4$ μ M, Figure 4a), and Tau binds with similar micromolar affinity to the (Hsp90/FKBP51) preformed complex ($K_d = 8$ μ M, Figure 4a). For the revised version of the manuscript, we now also determined the affinity between FKBP51 and Tau ($K_d = 45$ μ M, new Supplementary Figure 6a), which is significantly lower than for binding of Tau to Hsp90, in agreement with NMR titrations (Figure 4a and new Supplementary Figure 6b-c). Further support for the notion of a platforming role of Hsp90 came from SEC-MALS (please see New Figure/Data 3 in the previous response to referees): although Tau and FKBP51 bind weakly to each other according to fluorescence affinity measurements and NMR (new Supplementary Figure 6a), the Tau/FKBP51 complex was not stable on SEC even after long incubation times. Only when all three proteins were present, a complex (the ternary Hsp90/FKBP51/tau complex) was detected by SEC-MALS (Figure 4c). Furthermore, crosslinks between FKBP51 and Tau were only observed in the presence of Hsp90 (Supplementary Table 1d).

Please also note that we observed some functional and structural changes of Hsp90: In ATPase activity measurements, binding of FKBP51 reduced Hsp90's ATPase activity to a 48%, while in the ternary complex it decreased to 58% (Figure 1c). SAXS showed that the R_g increases from 7.30 nm in the binary Hsp90/FKBP51 complex, to 7.52 nm in the ternary complex (Figure 1b and Supplementary Figure 1b). In addition, we observed changes in PRE distributions when the ternary complex was formed (Figure 5 and Supplementary Figure 6). We therefore believe that the combined data support a platforming role of Hsp90.

Reviewer #3: 2. In the rebuttal, page 2 paragraph 1, authors say “(Fig. 4a) shows that Hsp90’s binding affinity for Tau slightly increases in the presence of FKBP51”. But actually, Fig. 4a shows that the K_d value of Hsp90/Tau ($4.2 \pm 1.3 \mu\text{M}$) is lower than (Hsp90+FKBP51)/Tau ($8.2 \pm 0.7 \mu\text{M}$), suggesting higher affinity for Tau in the absence of FKBP51.

Reply: Thanks for spotting this mistake (it should read “decreases in the presence of FKBP51”). We corrected it in the revised version of the manuscript (page 32).

Reviewer #3: 3. In the new figure/Data 2, the authors say “In addition, we also performed NMR titration experiments adding increasing equivalents of FKBP51 to a preformed Tau:Hsp90 (1:2) complex (see new figure 2). There is almost no change in Tau’s binding profile in the presence of increasing amounts of FKBP51, suggesting that Tau binds to Hsp90 more strongly than to FKBP51, which, at our experimental conditions, is bound to Hsp90 (based on the nanomolar binding affinity for the Hsp90:FKBP51 complex, Supplementary Fig. 5b)”. But the problem is, FKBP51 itself would decrease the intensity of Tau too! So the “no change in Tau’s binding profile” could be also caused by Tau bind to FKBP51, which compensates for the change caused by Hsp90. To support the authors’ conclusion, they need to find there are some specific residues of Tau that only change in Hsp90 binding or only in FKBP51 binding. Otherwise this supporting result is not very convincing to conclude that Tau preferably bind Hsp90 but not FKBP51 in the ternary complex. Similarly, I think Figure 4d has same problem, in which the authors suggest that “Chemical shift changes induced in Tau by the Hsp90/FKBP51 complex were however very similar to those observed with Hsp90 alone (Fig. 4d)”. But again, authors could not rule out that FKBP51 itself could affect Tau too, meaning that in the sample of Tau+Hsp90+FKBP51 (1:2:2) in Fig. 4d, some portion of Tau bind FKBP51 only and could also contribute to signal attenuation. Besides, if Tau bind to the Hsp90 in Hsp90/FKBP51 complex, we should expect the intensity change be larger than the Tau/Hsp90, because Hsp90/FKBP51 is a larger complex than Hsp90 alone, which will cause more signal reduction. But from Fig. 4d, Tau+Hsp90+FKBP51(1:2:2) is similar or even a little weaker than Tau+Hsp90(1:2).

Reply: We fully agree with the reviewer that interpretation of changes induced in NMR signal position and intensity is complicated due to their high sensitivity to a variety of parameters related to complex formation (K_d , structural changes induced upon binding, chemical shift changes between free and bound state). The new affinity data of the FKBP51/Tau interaction (New Supplementary Figure 6a) support the statement that Tau binds to Hsp90 more strongly than to FKBP51. Because analysis of the ternary complex in terms of K_d and K_{off} of individual components is complicated, we removed statements regarding different affinities within the ternary complex from the revised version of the manuscript.

Reviewer #3: 4. Authors say “chromatographic measurements demonstrated that FKBP51 stabilizes the ternary complex (Fig. 4b-c)” (line 215). But to me, the asterisk peaks, which are ternary complex suggested by the authors, contain mostly Hsp90 and FKBP51 but not Tau or K32 (I can’t find which band is Tau, is it completely overlapped with FKBP51?). I wonder what the chromatograph of Hsp90-FKBP51 looks like (I am surprised the authors did not show it. In the New Figure/Data 3 they show K32 and FKBP51, but I did not find any SEC of Hsp90 and FKBP51 complex). It could help to tell that the three samples chromatography are different from the 2 samples’ one, suggesting the existence of ternary complex.

Reply: To clarify this point, we performed additional SEC-MALS experiments:

SEC-MALS of the Hsp90/FKBP51 binary complex vs. the Tau/Hsp90/FKBP51 ternary complex. The binary complex was subjected to a Superdex 200 SEC column, the ternary complex to a Superose 6 SEC column, because of the high molecular weight peak. The lower panel corresponds to that shown in Figure 4c.

SEC-MALS of the Hsp90/FKBP51 binary complex showed only a very weak high-molecular weight peak compared to that found for the ternary complex, suggesting that - similar to what was shown for the Hsp90/Hop complex by the Agard group (ref #49 in our manuscript) - the Hsp90/FKBP51 largely dissociates in the SEC column.

Regarding the gel shown in Figure 4c: the band corresponding to Tau overlaps with that of FKBP51. This is now mentioned in the corresponding Figure legend (page 32). For this reason, we performed SEC-MALS also for the Hsp90/FKBP51/K32 complex (the band corresponding to K32 is labeled; please see above).

Reviewer #3: 5. In my opinion, the most significance of the ternary complex, is the existence of FKBP51 would introduce an orientational bias in the ensemble of Tau conformations in Hsp90 binding (“Targeting Tau to the catalytic center in the Hsp90/PPIase complex” in Results, line 278-296). This suggest how co-chaperone might remodel the Hsp90’s client binding. But this conclusion is only supported by one PRE experiment (178 of Tau). As they also labeled at 352 of Tau for testing interaction with Hsp90 (Supplementary Figure 6b), I wonder how this would look like when adding Hsp90/FKBP51 complex. Besides, authors say “In addition, Tau’s proline-rich region clusters around the catalytically active FK1 domain in the Hsp90/PPIase/client complex (Fig. 5d)”. I did not understand how the PRE result supports this. Moreover, does it contradict the authors’ other conclusion that Tau preferentially binds Hsp90 only in the ternary complex?

Reply: Please note that we labeled individually three different sites in Tau with MTSL (residues 178, 256 and 352) and determined the PRE broadening induced by these three spin-labeled Tau proteins in both the binary Hsp90/Tau and the ternary Hsp90/FKBP51/Tau complex (please see Supplementary Figure 7 and Supplementary Table 1e). In Fig. 5a and 5c, we illustrate the PRE broadening induced by Tau MTSL-labeled at 178, because for this position of the MTSL-label the most pronounced differences in Tau-induced PRE broadening of Hsp90

residues between the binary Hsp90/Tau and the ternary Hsp90/FKBP51/Tau complex were observed. Although the system is highly dynamic, we believe that the changes in PRE broadening (as particularly seen for 178-labeled Tau; Fig. 5) when going from the Hsp90/Tau to the Hsp90/FKBP51/Tau complex experimentally support the orientational bias introduced by FKBP51.

Residue 178 is located in the proline-rich region of Tau. In the binary Hsp90/Tau complex, the MTSL label in this position broadens residues in all three Hsp90 domains (including Hsp90-C; Fig. 5a and Supplementary Table 1e), while in the ternary complex only residues in Hsp90-N and Hsp90-M are broadened (Fig. 5c and Supplementary Table 1e). In addition, a smaller number of residues in Hsp90-M was broadened by MTSL-178-Tau in the ternary complex when compared to the Hsp90/Tau complex (Supplementary Table 1e). These data suggest that the proline-rich region of Tau preferentially locates (“clusters”) to the FK1 domain of Hsp90 in the ternary complex. We do not think that this contradicts our statement that Tau binds preferentially to Hsp90 in the ternary complex, because Hsp90 does not act just as an “inert” (that is not affected by FKBP51; please see our reply above) binding protein in the ternary complex.

Reviewer #3: Question 2: About structural model calculation and validation. I do not agree with the reviewer 2’s comment that “The Hsp90-FKBP51 complex was built on the basis of just (!) 7 inter-molecular NOEs. The interacting surface is very large and typically would require several hundred of NOEs to be reasonably defined.” I think when considering inter-molecular NOEs, the reliability is more important than the number. A few reliable inter-molecular NOEs combine with some other ambiguous restraints could be applied to get a very reasonable structural model. As it is a structural model, we could not ask for high accuracy at every residue. Rather, the further experimental validation would be important for testing the model and provide a solid conclusion.

Reply: We are happy that the referee shares our opinion on the importance of reliable inter-molecular NOEs and the subsequent validation of structural models.

Reviewer #3: There appears to be some problem or lack of clarity of the data in the model building: 1. In the Results (line 175-187) and Methods (line 609-641), the authors say they use unambiguous inter-molecular NOEs and ambiguous restraints from cross-links to calculate the model, and use PRE and PCS to validate the model. I think it is more reliable to add the PRE and PCS in the model calculation (at least final model calculation), especially the very decent PCS result. I think it would be very helpful for the convergence and clustering. I wonder how the model would look if it was calculated with just the inter-molecular NOEs, PRE and PCS.

Reply: When the manuscript was reviewed at Nature, the reviewers asked for additional validation of the calculated structural models. We therefore spent a lot of time in designing mutants of Hsp90 and FKBP51, which behave sufficiently well and where we could attach an MTSL (Hsp90) or lanthanide tag (FKBP51), followed by PRE/PCS measurements. The obtained PRE/PCS data provided strong support for the determined structural models (Fig. 3). If we introduce the PCSs and PREs as restraints into the structure calculation, we would lose this ability to validate the resulting structural models. Given the strong criticism, which we faced by reviewer 2, we think that this could be problematic. Please also note that PREs provide less stringent restraints than intermolecular NOEs (both provide distance information) and the PCSs fit very well to the structural models (Fig. 3f,g), suggesting that the inclusion of the PREs/PCSs as restraints would not strongly change the structural models. On the other hand, through the inclusion of the cross-link data (every cross-link experiment was repeated six

times and only reproducible crosslinks were included as restraints) into the structure calculation, we were able to cover a larger portion of the two proteins, since no sequence-specific assignments were obtained for the TPR region of FKBP51 (due to protein aggregation; please see the text on pages 12-13).

Reviewer #3: Question 2: 2. In the rebuttal, authors say the Hsp90-FKBP51 complex calculation are using data from following parts: • Residue-specific NMR chemical shift perturbation, • PCSs and PRE data, • 16 inter-molecular crosslinks, • 7 inter-molecular NOEs, • SAXS data. But I have some questions here: 1) about the “Residue-specific NMR chemical shift perturbation”, I did not see this mentioned elsewhere in the paper, either in Results or Discussion. What does the CSP look like?

Reply: Thanks for spotting the mistake. It should read “Residue-specific NMR signal perturbation” and refers to the changes in NMR signal intensity observed in the NMR titrations (Figure 1d, 2e and Supplementary Figure 2). In the NMR titrations, we observed predominantly NMR signal broadening. Please see below an overlay of CSP and signal broadening observed in ^{15}N -labeled Tau upon addition of Hsp90/FKBP51 (Tau/Hsp90/FKBP51 molar ratio of 1:2:2).

Superposition of signal broadening (in orange line, left axis) and CSP (dark red bars, right axis) in ^{15}N -labeled Tau upon addition of Hsp90/FKBP51 (Tau/Hsp90/FKBP51 molar ratio of 1:2:2). Please note the small scale used for the CSP axis.

Reviewer #3: 2) the authors mention here that they use PCSs and PRE data for calculation, but in the result and method part, they mention that PRE and PCS are used to validate the model. Which one is true? They cannot use PRE and PCSs for both calculation and validation. The common strategy is use the PRE and PCSs for model calculation. For validation, mutagenesis and truncation provide more solid result. The authors need to clarify the model structural calculation method.

Reply: We double checked that in the manuscript everything is accurately described (i.e. that PREs and PCSs were used for cross-validation and not for structure calculation; please see also above). In our reply to referee #2 (her/his comments #1 and #3) we also state that the PCSs were used for cross-validation, and we listed all NMR parameters, which were used either for structure calculation (intermolecular NOEs and cross-links) or validation (PREs, PCSs, NMR signal broadening, SAXS data; reply to comment #2 of referee #2). Maybe it was misleading that we wrote in our rebuttal to referee #2 “for building the complexes” and did not distinguish between structure calculation and validation. We apologize for the confusion.

Reviewer #3: Question 3: About some detail of Hsp90-FKBP51 interaction. 1. Since the authors could detect inter-molecular NOEs between Hsp90N/FK1, Hsp90CL/FK1, Hsp90M/FK1, Hsp90M/FK2, Hsp90M/FK2, at equimolar protein, it is puzzling that, in their titration experiment, FK1 and FK2 do not show significant signal attenuation even with 2 fold addition of Hsp90, but requires a 4 times excess of Hsp90. Could this attenuation cause by the nonspecific binding of FK1 and FK2 to more than one Hsp90 under these conditions of excess Hsp90? It is rather dubious to suggest the presence of specific cross-links, PREs and NOEs, for a weak nonspecific interaction.

Reply: At equimolar ratio, we observed an intensity decrease of on average ~20% in FK1 and ~25% in FK2 (and ~72% in TPR, please see Fig. 2g). At higher Hsp90 concentration, NMR signal intensities in FK1 and FK2 further decreased in parallel with the intensity decrease in the TPR domain (Fig. 2g). We clarified this point in the modified version of the manuscript (page 5). The differential binding kinetics of the domains are also in line with differences in the intensity of intermolecular NOEs – that depend with $1/r^6$ on the intermolecular distance and are thus sensitive to small populations of the complex – observed to different FKBP51 domains (please see Supplementary Figure 3a). For example, the NOE between Hsp90C and the TPR region (detected at equimolar ratio) is more intense than intermolecular NOEs that involved other regions of the proteins (Supplementary Figure 3a), suggesting that this specific contact is kinetically more favorable resulting in a stronger NOE cross peak. Moreover, the robust cross-validation of the structural model of the Hsp90/FKBP51 complex (and in particular of the FK1/FK2 vs Hsp90N/Hsp90M orientation) by the experimental PCSs (which are sensitive to both distance and orientation; Fig. 3), supports a specific interaction with FK1 and FK2. Please also note that we observed very little signal broadening (Supplementary Fig. 2d,e) at even 2-fold excess of FKBP51 over Hsp90-NM, i.e. an Hsp90 construct which lacks the TPR domain that is essential for formation of the specific/high-affinity Hsp90/FKBP51 complex.

Reviewer #3: Question 3: 2. In Fig. 2h, concerning the interaction of Hsp90 and FK2TPR, deleting FK1 domain mildly decreases the Hsp90 binding affinity (Supplementary figure 5) and weakly accelerates the binding rate (Fig. 2h). Is this because FK1 is present in bound and unbound conformations? Could it also be caused by the rotation of Hsp90N to accommodate FK1 in the complex as suggested by the authors (line 193-198)? About the rotation of Hsp90N in the complex structure, how reliable is the evidence of this? According to the Methods (line 623-641), the authors first calculate the Hsp90MCCM/FK1FK2 complex, and then Hsp90N is added after that. Could the observation be an artifact of the conditions of the calculation?

Reply: Because FK1 is present in bound and unbound conformations in the complex, deletion of FK1 decreases the interaction surface with Hsp90 and thus the enthalpy of the interaction upon saturation (Supplementary Figure 5). At the same time, deletion of FK1 accelerates the binding rate, because in the mechanism of complex formation illustrated in Figure 2, binding of FK1 is the last, and a potentially rate-limiting step.

Regarding the rotation of Hsp90N with respect to Hsp90M, the Mayer group (ref #35 in our manuscript) showed that Hsp90N can rotate up to 180° in solution compared to the crystal structures deposited in the PDB. In contrast, the relative orientation of FK1 and FK2 was kept the same as in the crystal structure of FKBP51, because NOE and RDC measurements had shown that FK1-FK2 behave as a largely rigid unit in solution. Based on these findings, we first docked FK1-FK2 to Hsp90M, followed by NOE/crosslink-based docking of Hsp90N. The conformation of Hsp90N-Hsp90M/FK1-FK2, which was obtained in this manner, was subsequently validated by PREs and PCSs (Fig. 3). Please note that we stress in the discussion, that the model of the Hsp90/FKBP51-complex shown in Fig. 3 is only representative of the fully

saturated condition/fully bound complex. Thus, at lower molar ratios Hsp90N can still rotate. This could also explain why - at equimolar concentration - FKBP51 decreased Hsp90's ATPase activity by only 50% (Figure 1c).

Reviewer #3: Some minor questions about analyzing the intensity profile. 1. For some NMR titrations, authors use very high concentration ratio. In Fig. 2e, the FK1 and FK2 domain of FKBP51 show intensity change after adding 4-5-fold of Hsp90. Considering the relative high concentration of sample they use (~100uM of labeled FKBP51), these change may be caused by unspecific binding and thus is not biological relevance.

Reply: Please refer to our detailed response to Question 3/point 1, i.e. the combined data and in particular the PCS data exclude unspecific binding, and the observed effects on ATPase activity support the biological relevance.

Reviewer #3: 2. In Fig. 4d and new Figure/Data 1, by comparing the intensity change of Tau with Hsp90 and FKBP51, the authors conclude that as the change of Tau with Hsp90 is more significant, Tau binding Hsp90 is stronger than with FKBP51. I think it is not very convincing because the difference of the signal attenuation in Tau could be caused by the different size and dynamics of Hsp90 and FKBP51. To carefully make this conclusion, authors should measure the Kd value of Tau and FKBP51, as reviewer 1 suggests.

Reply: We determined the affinity for binding of Tau to FKBP51 (new Supplementary Figure 6a). The Kd for this interaction is 45 μ M, while Tau binding to Hsp90 is about 10-fold stronger (Kd= 4 μ M, Figure 4a).

REVIEWERS' COMMENTS:

Reviewer #3 (Remarks to the Author):

In general, the revised manuscript seems ready for publication. A few questions remain:

1) The explanation about the Hsp90 "platforming role" for Tau and FKBP51 (page 1 of new rebuttal) is still not satisfactory. In the newest rebuttal, authors explain that: 1) the affinity of Hsp90 and FKBP51 is relatively strong, while Hsp90 to Tau and Tau to FKBP51 are much weaker; 2) the Tau/FKBP51 complex is too weak to stabilize, but the ternary complex is able to be detected by SEC-MALS; 3) the ternary complex decreases the Hsp90's ATPase activity more than just FKBP51; 4) SAXS shows a different Rg value for ternary complex; 5) change of PRE distributions (for Hsp90) in ternary complex.

But as I described in the original question, my doubt is that the description of authors would mislead readers that the presence of Hsp90 provides new binding features for FKBP51 and Tau, which is not the case as shown by the reported results that lead the authors to suggest that in the ternary complex, FKBP51 and Tau remain unbound to each other. It is probably more appropriate to propose that FKBP51 acts as a co-chaperone for Hsp90 binding to client (Tau).

2) For the question of PRE data (page 3-4 of the new rebuttal), about different MTSL labeled sites of Tau to, Hsp90 only and Hsp90/FKBP51 (Fig. 5 and Supplementary Fig. 7), how do the authors explain that proline-rich region of Tau binds with an orientational bias but the MBD does not show this bias? Could this observation have any functional relevance?

The authors also state "These data suggest that the proline-rich region of Tau preferentially locates ("clusters") to the FK1 domain of Hsp90 in the ternary complex", but the FK1 domain is FKBP51. This is actually what I meant in the original question, they are observing the spectrum of Hsp90, but why they explain it is effect to FK1 domain (which should be FKBP51)? Is it a mistake? Should it be FK1 domain-binding region of Hsp90?

3) For the SEC-MALS data (page 3 of the new rebuttal), it is obvious that the Hsp90/Tau/FKBP51 ternary complex looks different from the Hsp90/FKBP51 complex. But I am just curious that in the SDS-PAGE, the sample of Hsp90/K32/FKBP51 and sample of Hsp90/Tau/FKBP51 are actually very similar in 60 kDa region, how the authors could tell that there are Tau in the Hsp90/Tau/FKBP51 sample?

Reviewer #3 (Remarks to the Author):

In general, the revised manuscript seems ready for publication. A few questions remain:
1) The explanation about the Hsp90 “platforming role” for Tau and FKBP51 (page 1 of new rebuttal) is still not satisfactory. In the newest rebuttal, authors explain that: 1) the affinity of Hsp90 and FKBP51 is relatively strong, while Hsp90 to Tau and Tau to FKBP51 are much weaker; 2) the Tau/FKBP51 complex is too weak to stabilize, but the ternary complex is able to be detected by SEC-MALS; 3) the ternary complex decreases the Hsp90’s ATPase activity more than just FKBP51; 4) SAXS shows a different Rg value for ternary complex; 5) change of PRE distributions (for Hsp90) in ternary complex. But as I described in the original question, my doubt is that the description of authors would mislead readers that the presence of Hsp90 provides new binding features for FKBP51 and Tau, which is not the case as shown by the reported results that lead the authors to suggest that in the ternary complex, FKBP51 and Tau remain unbound to each other. It is probably more appropriate to propose that FKBP51 acts as a co-chaperone for Hsp90 binding to client (Tau).

Reply: The platforming role attributed to Hsp90 means that Hsp90 will promote the interaction of FKBP51 and Tau by providing different binding platforms on Hsp90’s structure, or by recruiting both proteins in close proximity (as affinities suggest). Therefore, Hsp90 will facilitate FKBP51’s PPIase activity on Tau’s proline rich region. Furthermore, FKBP51 bound to Hsp90 will promote additional structural changes on Tau once Tau is bound to the preformed Hsp90/FKBP51 complex, which goes in line with its co-chaperoning role for binding to the client, as the reviewer suggests. This platforming role for Hsp90 provides an explanation for the synergetic pro-toxic effect on Tau observed when the Hsp90/FKBP51 complex is formed (*i.e.*: when Hsp90 is included, FKBP51 effects on Tau aggregation are strongly enhanced) (Blair et al. J Clin Invest 123, 4158 (2013)). We are therefore convinced that the combined data provide substantial evidence to conclude that Hsp90 provides a platform for the simultaneous interaction of FKBP51 and Tau, and that Hsp90 facilitates the catalytic activity of FKBP51 on the pertinent region of Tau. However, we also revised the manuscript to reduce any possible misleading suggestions that the presence of Hsp90 provides new binding features for FKBP51 and Tau. As suggested by the reviewer, we also stress the co-chaperone role of FKBP51 in the complex.

2) For the question of PRE data (page 3-4 of the new rebuttal), about different MTSL labeled sites of Tau to, Hsp90 only and Hsp90/FKBP51 (Fig. 5 and Supplementary Fig. 7), how do the authors explain that proline-rich region of Tau binds with an orientational bias but the MBD does not show this bias? Could this observation have any functional relevance? The authors also state “These data suggest that the proline-rich region of Tau preferentially locates (“clusters”) to the FK1 domain of Hsp90 in the ternary complex”, but the FK1 domain is FKBP51. This is actually what I meant in the original question, they are observing the spectrum of Hsp90, but why they explain it is effect to FK1 domain (which should be FKBP51)? Is it a mistake? Should it be FK1 domain-binding region of Hsp90?

Reply: For the acquisition of the PRE data, we have performed the following 6 experiments:

- a) MTSL-labeled Tau on position 178 binding to Ile-labeled Hsp90.
- b) MTSL-labeled Tau on position 178 binding to the complex formed by (Ile-labeled Hsp90 + Leu/Val-labeled FKBP51).
- c) MTSL-labeled Tau on position 256 binding to Ile-labeled Hsp90.
- d) MTSL-labeled Tau on position 256 binding to the complex formed by (Ile-labeled Hsp90 + Leu/Val-labeled FKBP51).
- e) MTSL-labeled Tau on position 352 binding to Ile-labeled Hsp90.

- f) MTSL-labeled Tau on position 352 binding to the complex formed by (Ile-labeled Hsp90 + Leu/Val-labeled FKBP51).

Thus, for the samples where only Hsp90 is labeled (experiments a,c,e from the list), we could only detect PREs on Hsp90 moieties. However, in samples where both Hsp90 and FKBP51 were labeled (b,d,f) PREs on both proteins can be detected upon formation of the ternary complex. This allowed us to observe changes on PRE distributions on Hsp90 upon addition of FKBP51 (because we observed signals from Hsp90), as well as PREs on FKBP51 (because we also observed signals from FKBP51 in the spectra). Since we used different labeling strategies (Ile for Hsp90 and Leu/Val for FKBP51), we unambiguously observed PREs on both proteins. In addition, since we have labeled Tau at three different positions, we mapped the ensemble of different regions of Tau, and how it changes in the ternary complex. We have carefully checked the manuscript to avoid any misinterpretation of the data.

As explained in Supplementary Figure 7, we did not detect changes on Tau's MBD when binding to Hsp90 and to the Hsp90/FKBP51 complex, but only on Tau's proline-rich region. Since there is a connection between proline isomerization, hyperphosphorylation and aggregation in Tau, as we state in the manuscript (page #12), we conclude that the clustering of Tau's proline-rich region around FK1 is functionally relevant.

3) For the SEC-MALS data (page 3 of the new rebuttal), it is obvious that the Hsp90/Tau/FKBP51 ternary complex looks different from the Hsp90/FKBP51 complex. But I am just curious that in the SDS-PAGE, the sample of Hsp90/K32/FKBP51 and sample of Hsp90/Tau/FKBP51 are actually very similar in 60 kDa region, how the authors could tell that there are Tau in the Hsp90/Tau/FKBP51 sample?

Reply: Because Tau and FKBP51 co-migrate in SDS-PAGE and their bands overlap, we decided to perform the experiment also using a shorter fragment of Tau (K32) that contains all the relevant regions of Tau for the interaction. Because of the smaller size of K32, it will not overlap with FKBP51 in the gel (please see the Figure legend). In this way, we unambiguously detected the three bands in the gel and thus the presence of the three proteins in the elution profile for the ternary complex. Please note that the corresponding K32 band shows somehow lower intensity, which could explain why there is not a great change in the region around 60 kDa for the Hsp90/Tau/FKBP51 sample, as the reviewer points out.